# Unsupervised Panoptic Interpretation of Latent Spaces in GANs Using Space-Filling Vector Quantization

**Mohammad Hassan Vali**                                                        *mohammad.vali@aalto.fi*
*Department of Computer Science, Aalto University*

**Tom Bäckström**                                                              *tom.backstrom@aalto.fi*
*Department of Information and Communications Engineering, Aalto University*

**Reviewed on OpenReview:** *https://openreview.net/forum?id=SEJatSGZX8*

## Abstract

Generative adversarial networks (GANs) learn a latent space whose samples can be mapped to real-world images. Such latent spaces are difficult to interpret. Some earlier supervised methods aim to create an interpretable latent space or discover interpretable directions, which requires exploiting data labels or annotated synthesized samples for training. However, we propose using a modification of vector quantization called space-filling vector quantization (SFVQ), which quantizes the data on a piece-wise linear curve. SFVQ can capture the underlying morphological structure of the latent space, making it interpretable. We apply this technique to model the latent space of pre-trained StyleGAN2 and BigGAN networks on various datasets. Our experiments show that the SFVQ curve yields a general interpretable model of the latent space such that it determines which parts of the latent space correspond to specific generative factors. Furthermore, we demonstrate that each line of the SFVQ curve can potentially refer to an interpretable direction for applying intelligible image transformations. We also demonstrate that the points located on an SFVQ line can be used for controllable data augmentation.

## 1 Introduction

Generative adversarial networks (GANs) (Goodfellow et al., 2014) are powerful generative models applied to various applications, e.g., data augmentation (Antoniou et al., 2017; Shorten & Khoshgoftaar, 2019), image editing (Härkönen et al., 2020; Yüksel et al., 2021; Shen & Zhou, 2021; Voynov & Babenko, 2020; Tzelepis et al., 2021; Aoshima & Matsubara, 2023; Abdal et al., 2021; Wang et al., 2018b; Alaluf et al., 2022; Roich et al., 2022; Pehlivan et al., 2023; Liu et al., 2023; Jahanian et al., 2019; Plumerault et al., 2020; Yang et al., 2021; Goetschalckx et al., 2019; Shen et al., 2020; Wu et al., 2021), and video generation (Wang et al., 2018a). For image data, GANs map a latent space to an output image space by learning a non-linear mapping (Voynov & Babenko, 2020). After learning such mapping, GANs can create realistic high-resolution images by sampling from the latent space (Karras et al., 2019). However, this latent space is a black box, making it difficult to interpret the mapping between the latent space and generative factors such

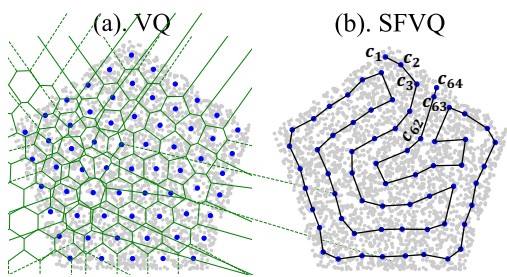

Figure 1: Codebook vectors (*blue points*) of a 6 bit (a) vector quantization, and (b) space-filling vector quantization (*curve in black*) on a pentagon distribution (*gray points*). Voronoi regions for VQ are shown *in green*.

as gender and age (Shen et al., 2020). In addition, the interpretable directions to change these factors are not known (Voynov & Babenko, 2020). Hence, having a comprehensive interpretation of the latent space is an important research problem that, if solved, leads to more controllable generations.

In the literature, both supervised and unsupervised methods exist to find interpretable directions in the latent space. Supervised methods (Jahanian et al., 2019; Plumerault et al., 2020; Yang et al., 2021; Goetschalckx et al., 2019; Shen et al., 2020; Wu et al., 2021; Shen et al., 2022) require data collections together with the use of pre-trained classifiers or human labelers to label the collected data with respect to the user-predefined directions (Shen & Zhou, 2021). In addition, these methods only find the directions that the user defines (Voynov & Babenko, 2020). On the other hand, in unsupervised methods (Härkönen et al., 2020; Shen & Zhou, 2021; Voynov & Babenko, 2020; Yüksel et al., 2021; Tzelepis et al., 2021; Aoshima & Matsubara, 2023; Song et al., 2023b;a), the user has to choose the hyper-parameter $K$ (the number of interpretable directions to discover) before training, where a large value for $K$ results in discovering repetitive directions (Yüksel et al., 2021). In all these unsupervised methods, there is no prior knowledge about the specific transformation each of these $K$ discovered directions yields. Hence, the user should do an exhaustive search over all available $K$ directions to determine which directions are practical and what they refer to. For instance, as GANSpace (Härkönen et al., 2020) applies principal component analysis (PCA) on the latent space, the number of directions ($K$) to be examined is large (equal to the latent space dimension). Additionally, as stated in Härkönen et al. (2020), not all PCA directions are necessarily useful for changing a generative factor.

In this paper, we use a modification of vector quantization, called space-filling vector quantization (SFVQ) (Vali & Bäckström, 2023), to interpret the latent spaces of the pre-trained StyleGAN2 (Karras et al., 2020) and BigGAN (Brock et al., 2018) models using FFHQ, AFHQ, LSUN Cars, CIFAR10, and ImageNet datasets. Regarding the intrinsic arrangement of SFVQ codebook vectors (or codewords), index-wise adjacent codewords model the same locality of a distribution (see Fig. 1(b)). Hence, SFVQ can be used to capture the underlying structure of the GANs' latent spaces, given that subsequent codewords refer to similar contents. In contrast to supervised approaches, our unsupervised method neither requires human labeling nor imposes any constraints on the learned latent space, as it uses the original learned latent spaces from pre-trained models. Moreover, our method does not need any hyper-parameter tuning, e.g., choosing the number of directions ($K$) as in Shen & Zhou (2021); Voynov & Babenko (2020); Yüksel et al. (2021); Tzelepis et al. (2021); Aoshima & Matsubara (2023); Song et al. (2023b;a) or tuning the coefficients of the training loss terms as in Voynov & Babenko (2020); Yüksel et al. (2021); Tzelepis et al. (2021); Aoshima & Matsubara (2023). In our proposed method, to explore the latent space structure and find its interpretable directions, the only required effort is that the user should visually observe the generated images from the learned SFVQ codebook vectors (Fig. 6(a), Fig. 8(a)) only once. By observing the generated images, the user would have prior knowledge of the potential edit type for a discovered direction in advance, contrary to other unsupervised methods. Therefore, we reduce the search effort to achieve the desired edit by only searching for the suitable layers in StyleGAN2 or BigGAN to modify. Our method implementation is publicly available at https://github.com/Speech-Interaction-Technology-Aalto-U/Interpretable-GANs-by-SFVQ.git.

In Vali & Bäckström (2023), SFVQ training is prone to outlier codewords, i.e., there might be some codebook vectors that end up outside the data distribution, which is an issue. In this paper, we resolve this issue by improving the *initialization* (Sec. 3.2.1) and *codebook expansion* (Sec. 3.2.2) procedures of SFVQ such that we do not encounter any outlier codewords throughout our experiments. We show that our trained SFVQ codebook (or curve) can capture a universal interpretation of the StyleGAN2's latent space such that the user can identify what type of generations to expect from each part of the latent space regarding age, gender, pose, accessories for FFHQ, color, breed, pose for AFHQ, and class of data for CIFAR10 (see Sec. 5.1). Furthermore, we explore SFVQ from a new viewpoint and discover that SFVQ lines (the lines connecting SFVQ's subsequent codewords) can refer to interpretable directions leading to meaningful image transformations (see Sec. 5.2 and Sec. 5.3). This SFVQ property was not explored in Vali & Bäckström (2023). Qualitative (Sec. 5.4) and Quantitative (Sec. 5.5) evaluations show that the discovered interpretable directions by our proposed method outperform the directions of GANSpace (Härkönen et al., 2020), LatentCLR (Yüksel et al., 2021), and SeFa (Shen & Zhou, 2021) methods. In addition, our proposed method is capable of finding joint interpretable directions that can change multiple attributes simultaneously (see Sec. 5.7). By improving the SFVQ training, we observe that the learned SFVQ curve (or space-filling lines) are mainly located inside the latent space. Hence, we have a large number of meaningful latent vectors located on the SFVQ curve that can be used for controllable data augmentation. We observe that by sampling latent vectors from the line connecting two subsequent codewords, we can generate images that visually share the attributes of those codewords (see Sec. 5.8).

## 2 Related Work

Prior works can be categorized into three principal approaches that aim to make the latent space of generative models more interpretable.

**1. Introducing structure into the latent space using data labels**. The main rationale behind these approaches (Klys et al., 2018; Xue et al., 2019; An et al., 2021) is that they take advantage of labeled data (with respect to the features of interest) and train the generative model in a supervised manner to learn a structured latent space in which data with specific labels reside in isolated subspaces of the latent distribution. Hence, this structured latent space can be interpretable, allowing the user to have control over data generation and manipulation with respect to the labels. However, these supervised methods suffer from two main drawbacks. First, they require human labeling, whose cost can increase excessively as the dataset size increases (Voynov & Babenko, 2020). Second, they might prevent the latent space from learning some intrinsic structures that a human labeler is unaware of (Voynov & Babenko, 2020).

**2. Disentangling the latent space dimensions**. These methods (Chen et al., 2016; Higgins et al., 2017; Ramesh et al., 2018; Lee et al., 2020; Liu et al., 2020) train the generative model in an unsupervised way to obtain a disentangled latent space. In a disentangled latent space, changes in each latent dimension make variations only in one specific generative factor while keeping the other generative factors unchanged. In other words, these approaches aim to model various generative factors existing in the data to different latent dimensions and make these dimensions (generative factors) independent of each other. Therefore, these methods are interpretable in that they allow control over data generation with respect to the generative factors. However, the downside of these techniques is their low efficiency in generating quality and diversity (Voynov & Babenko, 2020).

**3. Exploring interpretable directions in the latent space**. The main goal of these techniques is to find the directions in the latent space which lead to intelligible data transformations such as changing the age, pose, hairstyle in face synthesis task (Härkönen et al., 2020; Voynov & Babenko, 2020; Shen et al., 2020; Jahanian et al., 2019; Yüksel et al., 2021; Shen & Zhou, 2021; Abdal et al., 2021; Wang et al., 2018b; Plumerault et al., 2020; Yang et al., 2021; Goetschalckx et al., 2019; Alaluf et al., 2022; Roich et al., 2022; Pehlivan et al., 2023; Liu et al., 2023; Tzelepis et al., 2021; Aoshima & Matsubara, 2023). Here, the interpretability of the latent space refers to the user's control over the generation process by manipulating latent vectors along these discovered interpretable directions. This category is the main focus of this paper, as we also use our proposed method to discover interpretable directions and compare these directions with those of other methods in the literature.

Some supervised methods (Jahanian et al., 2019; Plumerault et al., 2020; Yang et al., 2021; Goetschalckx et al., 2019; Shen et al., 2020; Wu et al., 2021; Shen et al., 2022) find interpretable directions guided by human supervision or pre-trained classifiers used to label the directions of the edited images, which are going to be used for training. For instance, InterFaceGAN (Shen et al., 2020) extracts a large set of latent vectors from the latent spaces of pre-trained PGGAN (Karras et al.) and StyleGAN (Karras et al., 2019). It used SVM to find the hyperplanes of the attributes, such that the normal vector of each hyperplane is considered the interpretable direction. To label the generated images along interpretable directions, InterFaceGAN uses pre-trained classifiers. The downside of these supervised methods is that they require sufficient data collection to discover convincing, interpretable directions. Some methods bypass these hassles by discovering interpretable directions in an unsupervised manner from the latent space of pre-trained models, without requiring training of any modules. GANSpace(Härkönen et al., 2020) applied principal component analysis (PCA) on the latent spaces of StyleGAN, StyleGAN2, and BigGAN and found that PCA directions can be used as interpretable directions. Similarly, in Shen & Zhou (2021); Song et al. (2023b), the eigenvectors of the affine transformation of the pre-trained model weights are considered interpretable directions. In a somewhat different technique, methods of Voynov & Babenko (2020); Yüksel et al. (2021); Yang et al. (2021); Tzelepis et al. (2021); Aoshima & Matsubara (2023) train a transformation function $f$ that shifts a latent vector $z$ in various directions and use a pre-trained generator $\mathcal{G}$ to generate images (or new latent vectors $w$) corresponding to the original ($z$) and shifted latent vector ($f(z) = z + \alpha \cdot \vec{d}$). Then, a reconstructor $\mathcal{R}$ (or attribute assessor) is trained to estimate or rate the direction $\vec{d}$ proposed by the transformation function $f$ given the pair of generated images (or latent vectors). When trained, the directions $\vec{d}$ are considered as

interpretable directions. In contrast to methods of Voynov & Babenko (2020); Yüksel et al. (2021); Yang et al. (2021) which only find linear directions, methods of Tzelepis et al. (2021); Aoshima & Matsubara (2023) are supposed to find non-linear directions.

GAN inversion methods (Alaluf et al., 2022; Liu et al., 2023; Katsumata et al., 2024; Dere et al., 2024; Zhu et al., 2024) revert the generation process by mapping real images to the latent space of pre-trained GANs. They take a real image as input and learn to find a latent vector for it such that the pre-trained GAN generator can reconstruct an image as close to the real image as possible. After computing the new latent space $\mathcal{W}_{inv}$ of pre-trained GANs, these methods mainly discover interpretable directions from $\mathcal{W}_{inv}$ using InterFaceGAN (Shen et al., 2020) and GANSpace (Härkönen et al., 2020) techniques. All the above-mentioned methods provide interpretable directions for image editing tasks in the latent space of GANs, such as BigGAN, PGGAN, and various versions of StyleGAN. However, in a completely different case, recent approaches have emerged for image editing in diffusion models (Park et al., 2023; Zhang et al., 2023; Dalva & Yanardag, 2024; Haas et al., 2024; Shi et al., 2024; Chen et al., 2024; Joseph et al., 2024; Sajnani et al., 2025; Feng et al., 2025). This paper does not focus on image editing using diffusion models; instead, it concentrates only on interpreting the latent space of the StyleGAN2 and BigGAN pre-trained models.

# 3 Methods

## 3.1 Space-filling vector quantization (SFVQ)

A space-filling curve is a piece-wise continuous curve created by recursion, and if the recursion repeats infinitely, the curve fills a multi-dimensional space (Sagan, 2012). Fig. 2 shows the first six recursion steps of the Hilbert curve (a well-known space-filling curve) that fills a square space. The curve is color-coded, i.e., it starts in light and ends in dark colors. It is clearly shown that there is an inherent structure

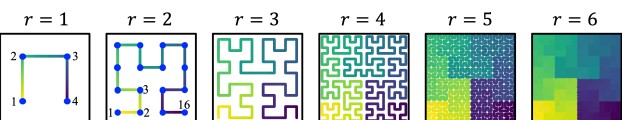

Figure 2: First six recursion steps ($r$) of a Hilbert space-filling curve. The curve is color-coded, i.e., it starts in light and ends in dark colors. Corner points are shown as *blue points*.

in the Hilbert curve and a good arrangement in the position of its corner points (shown *in blue*). For instance, adjacent corner points refer to the same locality of the space, as they are shown with similar colors. Motivated by space-filling curves, space-filling vector quantization (SFVQ) (Vali & Bäckström, 2023) designs vector quantization (VQ) as a mapping of data on a space-filling curve, whose corner points are the codebook vectors of a VQ process. SFVQ uses a dithering technique for training its codebook, i.e., it maps the input vector onto the line connecting subsequent codewords, but not necessarily onto the codewords. For an input vector $x \in \mathbb{R}^{1 \times D}$ and a codebook $C = \{c_1, \cdots, c_N\} \in \mathbb{R}^{N \times D}$ containing $N$ codewords, SFVQ first generates a dithered codebook matrix $C^d = \{c_1^d, \cdots, c_{N-1}^d\} \in \mathbb{R}^{N-1 \times D}$ by interpolation at random places on the line connecting two subsequent codewords of the codebook $C$ (see Fig. 3). Then, it quantizes (or maps) $x$ to the closest element from the dithered codebook $C^d$ as

$$\hat{x} = \arg\min_{c_i^d} \|x - c_i^d\|_2 \; ; \; 1 \leq i \leq N-1 \quad \Rightarrow \quad \hat{x} = c_j^d = (1-\lambda)c_j + \lambda c_{j+1}, \tag{1}$$

where $c_j$ and $c_{j+1}$ represent two subsequent codewords from the base codebook $C$ which their interpolation $c_j^d$ is the the closest dithered codeword to $x$, and $\lambda$ is the dithering (or interpolation) factor. To generate the dithered codebook during training (Fig. 3), SFVQ samples $\lambda$ values from the uniform distribution of $U(0,1)$ that ensures random interpolations between subsequent vectors of the base codebook $C$. When sampling different $\lambda$ values for different training batches, this type of randomized interpolation imposes a sense of continuity between subsequent vectors of $C$. Because the codebook $C$ is trained such that the line connecting its subsequent codewords should be a valid quantization point. The mean squared error (MSE) between the input vector $x$ and its quantized form $\hat{x}$ is used as the training loss function

$$\text{MSE}(x, \hat{x}) = \|x - \hat{x}\|_2^2 = \|x - (1-\lambda)c_j - \lambda c_{j+1}\|_2^2 \quad . \tag{2}$$

Similar to space-filling curves, SFVQ is trained recursively. SFVQ first starts with $N = 4$ codewords (2 bit) and after training these codewords for a while, it expands the codebook by doubling the number of codewords at each recursion step. The recursion continues until SFVQ reaches $N = \log_2(B_{target})$ codewords, where $B_{target}$ is the SFVQ target bitrate. Fig. 1 illustrates a 6 bit ($N = 64$) VQ and SFVQ applied on a 2D pentagon distribution. To clarify more, both VQ and SFVQ have a codebook containing $N = 64$ codewords. However, the codewords of SFVQ are in clear arrangement because their index-wise adjacent codewords refer to the same locality of the distribution, and also the line connecting subsequent codewords is valid for quantization. Whereas, these two properties do not hold for VQ (in general, not in this uniform pentagon 2D distribution).

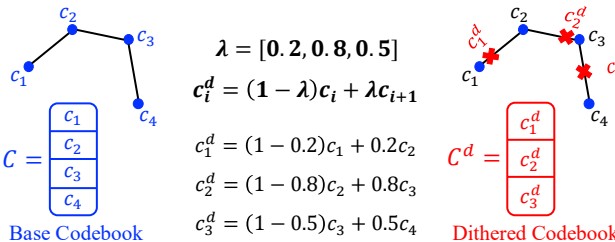

Figure 3: Example of generating dithered codebook $C^d$ from the base codebook $C$ for one typical training batch of space-filling vector quantization (SFVQ) in its first recursion step, where SFVQ starts with $N = 4$ codewords.

### 3.2 Proposed method

#### 3.2.1 SFVQ initialization

As mentioned, SFVQ training starts with $N = 4$ codewords. The *initialization* of these four codewords significantly impacts the final learned SFVQ curve obtained at the end of training. In Vali & Bäckström (2023), these codewords were initialized randomly (from a normal distribution $\mathcal{N}(0,1)$), and as the SFVQ codebook was expanded to reach the target bitrate, there were some unfavorable jumps (lines outside the distribution or lines breaking the codebook arrangement) in the learned curve (see Fig. 4(a)).

To address this issue, in this paper, we change the codebook *initialization*. Since pre-trained models are available, we sample $10^3$ random vectors $z$ from the normal distribution and generate their corresponding

(a) Random Initialization    (b) Proposed Initialization

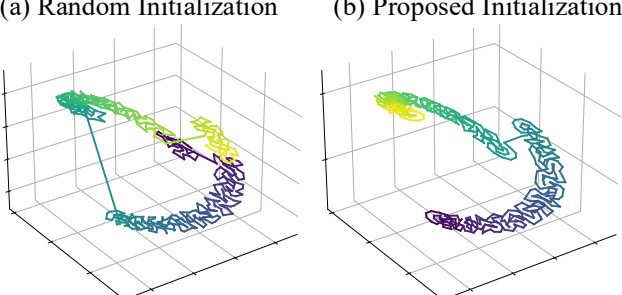

Figure 4: Two learned SFVQ curves with $N = 512$ codewords on a 3D *moon* dataset using random and our proposed initialization methods. The curves are color-coded, i.e., they start in light and end in dark colors.

latent vectors in the layer where we intend to train the SFVQ (e.g., intermediate $\mathcal{W}$ space in StyleGAN2). Then, we compute the Euclidean norm ($\ell_2$) of all latent vectors and sort them in ascending order. We split these sorted latent vectors into four groups and initialize the codebook vectors with the mean of these four groups. From a geometrical viewpoint, our proposed initial SFVQ curve spans from one end of the Euclidean latent space to its other end and brings a desirable order to SFVQ codewords, which aligns with the intrinsic SFVQ codebook arrangement, as the curve starts from low norm to high norm latent vectors. To confirm this fact, Fig. 4 shows two learned SFVQ curves (with $N = 512$ codewords) using random and our proposed initialization methods on a 3D *moon* dataset. The SFVQ curves are color-coded, i.e., they start in light and end in dark colors. As shown, our proposed initialization results in a perfect codebook arrangement with only one jump (which is inevitable as the curve should pass to the other cluster of the data). Whereas, the random initialization causes several unfavorable jumps that break the codebook arrangement.

#### 3.2.2 SFVQ codebook expansion

When training SFVQ, *codebook expansion* occurs at the beginning of each recursion step by doubling the codebook size. In Vali & Bäckström (2023), the new codebook vectors are defined in the center of the line connecting two adjacent codewords (which already exist on the curve), i.e., $c_{new} = (c_i + c_{i+1})/2$, where $c_i$ is the $i$-th codeword. $c_{new}$ can be useless if it is located outside the latent space, as in the case of undesirable

jumps (as shown in Fig. 4(a)), and it takes a long time (or many training batches) to be pushed inside. As discussed in Sec. 3.2.1, Vali & Bäckström (2023) uses random initialization that incurs unfavorable jumps, and hence, it can result in creating new codewords ($c_{new}$) out of the latent distribution in the *codebook expansion* step. To address this issue, in this paper, we define the new codeword by shifting the existing codebook vectors slightly such that $c_{new} = 0.99\,c_i + 0.01\,c_{i+1}$. Now, the new codewords most likely reside inside the latent space, and thus, after being selected actively during training, they will be optimized to their optimum locations. In contrast to Vali & Bäckström (2023), our proposed *codebook expansion* and *initialization* for SFVQ lead to no outlier codewords throughout our experiments.

### 3.2.3 Why are SFVQ lines likely to refer to interpretable directions?

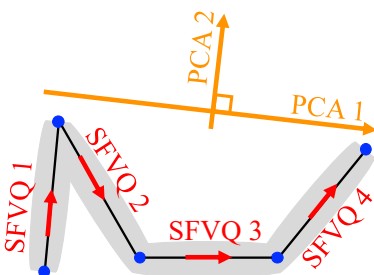

After training SFVQ with only $N = 4$ codewords on $\mathcal{W}$ space of Style-GAN2 pre-trained on FFHQ dataset, the obtained images from learned codewords are shown in Fig. 8(b). From the images, it can be inferred that SFVQ lines refer to two directions of gender and rotation (see Fig. 8(c)). Fig. 8(b) and Fig. 8(c) remind us of the PCA-based method of GANSpace (Härkönen et al., 2020) that finds PCA directions as interpretable directions. Similar to the first two PCA directions of GANSpace, which refer to changes in gender and rotation, the SFVQ lines are also located along the directions in which the training data has the most variance, i.e., gender and rotation. The reason is that SFVQ trains the codebook such that the points on the line connecting subsequent codewords should be valid quantization points. Hence, to minimize the MSE loss (Eq. (2)), SFVQ locates the codewords to some corners of the distribution so that their connecting lines lie along the directions that the distribution has the highest variances (see Fig. 5).

Figure 5: Directions found by SFVQ technique (*in red*) and PCA-based method of GANSpace (*in orange*) on an example distribution (shown *in gray*). The SFVQ curve is shown *in black* with its codewords shown *in blue*.

Fig. 5 shows directions found by two methods of SFVQ and GANSpace on an example distribution (shown *in gray*). According to the figure, the number of directions that GANSpace can find is restricted to the data dimension (in this case $D = 2$). In contrast, SFVQ can potentially find $N - 1$ distinct directions where $N$ is the number of codewords. Furthermore, the PCA directions of GANSpace are constrained to be orthogonal, meaning that they are perpendicular to each other. However, in the pre-trained $\mathcal{W}$ space of StyleGAN2, the interpretable directions are not necessarily orthogonal to each other, and that is why only the first 100 (out of 512) GANSpace's PCA orthogonal directions lead to noticeable changes (Härkönen et al., 2020). In contrast, in the SFVQ curve, the directions do not have an orthogonality constraint, and thus, each direction (or each SFVQ line) can potentially work for a meaningful and obvious change (see Fig. 5). Therefore, with regard to interpretable directions, SFVQ is somewhat similar to the PCA technique, but it has more degrees of freedom. Because SFVQ directions do not have the orthogonality constraint, SFVQ can potentially find $N - 1$ distinct meaningful directions. These observations and discussions motivate us to use the SFVQ curve to discover interpretable directions, which we study in Sec. 5.2.

### 3.2.4 How to find SFVQ interpretable directions?

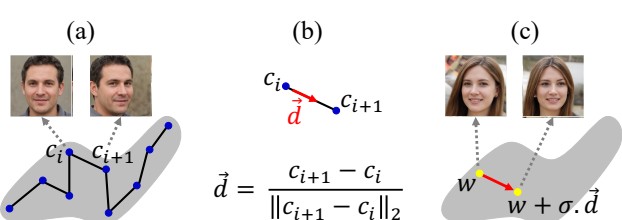

After training SFVQ on the latent space and obtaining the learned codebook, we generate the images corresponding to SFVQ codewords (Fig. 6(a) and similarly Fig. 7(a), Fig. 8(a)). This visualization reveals the underlying structure of the latent space in terms of generative factors, which we refer to as the *universal interpretation* of the latent space (Sec. 5.1). Similar to space-filling curves (Fig. 2), adjacent codewords of SFVQ refer to similar contents in the latent space. Hence, the learned SFVQ codebook (with its intrinsic arrangement) can capture the underlying structure of the latent space.

Figure 6: (a) Observation of generated images from learned SFVQ codebook (similar to Fig. 8(a)) to find a sensible direction between $c_i$ and $c_{i+1}$. (b) Computation of the direction $\vec{d}$. (c) Applying the direction on a random latent vector $w$ by shift magnitude of $\sigma$.

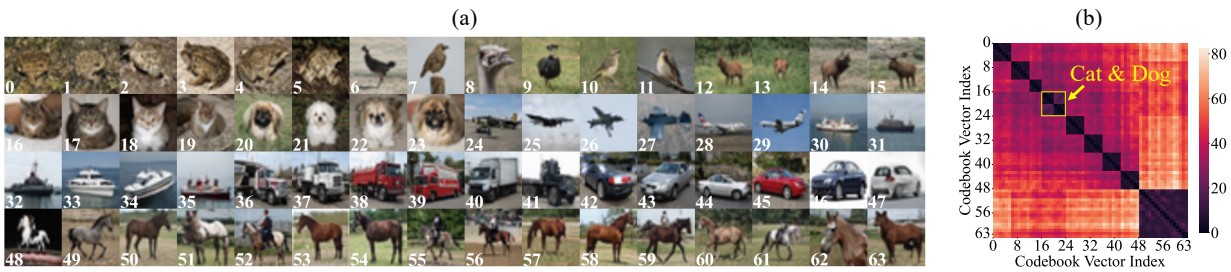

Figure 7: (a) Generated images from the codebook of a 6 bit SFVQ ($N = 64$) trained on $\mathcal{W}$ space of StyleGAN2 pre-trained on the CIFAR10 dataset. (b) Heatmap of Euclidean distances between all codebook vectors.

In this paper, we take a step forward in extracting more information from the SFVQ codebook, which results in finding interpretable directions. Fig. 6 shows how we find the interpretable directions using SFVQ. Due to the intrinsic arrangement of SFVQ codebook vectors, subsequent images refer to similar content, i.e., they share many similar features while differing in minimal attributes. For example, in Fig. 6(a), the images for two subsequent codewords of $c_i$ and $c_{i+1}$ share most of the attributes except the rotation. Hence, we can infer that the direction ($\vec{d}$) connecting these two codewords (Fig. 6(b)) refers to the rotation direction. Then, by shifting any latent vector along this direction (Fig. 6(c)), we can observe the change in rotation attribute for that latent vector.

By a quick observation of the subsequently generated images from the SFVQ codebook (Fig. 6(a)), the user can readily spot the interpretable direction. Hence, the user has prior knowledge of the direction, and the only required action is to find the proper layers of the GAN to edit along this direction (Härkönen et al., 2020). In this way, the user achieves the desired edit with less search effort compared to other unsupervised methods (Härkönen et al., 2020; Shen & Zhou, 2021; Voynov & Babenko, 2020; Yüksel et al., 2021; Tzelepis et al., 2021; Aoshima & Matsubara, 2023), in which apart from the layer-wise search, they should do an exhaustive search over all $K$ discovered directions to inspect whether they are practical and what directions they refer to.

## 4    Experiments

To evaluate how SFVQ can be used to interpret the latent spaces in GANs, similarly to GANSpace (Härkönen et al., 2020), we chose the intermediate latent space ($\mathcal{W}$) of StyleGAN2 (Karras et al., 2020) and the first linear layer of BigGAN512-deep (Brock et al., 2018), and then trained the SFVQ on these layers. These layers are more favorable for interpretation because they render more disentangled representations, are not constrained to any specific distribution, and suitably model the structure of real data (Karras et al., 2019; Härkönen et al., 2020; Shen et al., 2020). For StyleGAN2, we employ the pre-trained models on FFHQ (Karras et al., 2019), AFHQ (Choi et al., 2020), LSUN Cars (Yu et al., 2015), CIFAR10 (Krizhevsky et al., 2009) datasets, and also the pre-trained BigGAN on ImageNet datset (Deng et al., 2009).

We trained the SFVQ with various bitrates ranging from 2 to 12 bit ($N = 4$ to $N = 4096$ codebook vectors). Since the training of SFVQ is not sensitive to hyper-parameter tuning, we adopt a general setup that works for all pre-trained models and datasets. In this setup, we trained SFVQ with a batch size of 64 over 100 k number of training batches (for each recursion step) using the Adam optimizer with the initial learning rate of $1e^{-3}$. We used a learning rate scheduler such that during each recursion step, we halve the learning rate after 60 k and 80 k training batches. To show that SFVQ and its interpretation ability are not sensitive to the training hyper-parameters, we trained the SFVQ on the intermediate latent space ($\mathcal{W}$) of StyleGAN2 pre-trained on CIFAR10 dataset over different SFVQ bitrates, batch sizes, and learning rates. The results are provided in Appendix A.8.

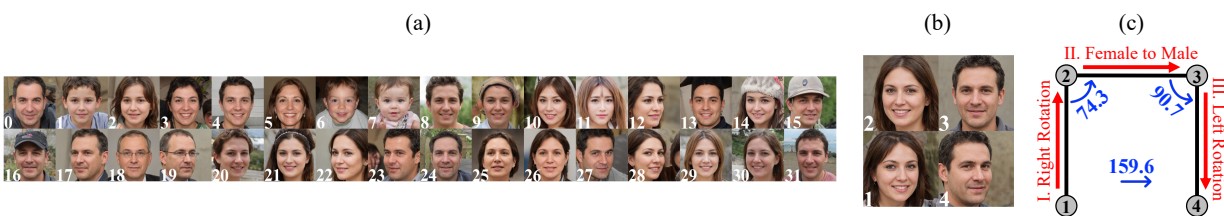

Figure 8: (a) Generated images from the codebook of a 5 bit SFVQ ($N = 32$) trained on $\mathcal{W}$ space of StyleGAN2 pre-trained on the FFHQ dataset. (b) Similar generated images for a 2 bit SFVQ ($N = 4$), and (c) their semantic directions. Numbers (*in blue*) show the angle between directions.

## 5 Results and discussions

### 5.1 StyleGAN2: universal interpretation

To explore a universal interpretation of the latent space, we apply the SFVQ on the latent space and plot the generated images from the obtained SFVQ codebook vectors. According to the inherent arrangement of the SFVQ codebook (see Sec. 3.1), adjacent codewords refer to similar contents of the latent space. Therefore, we expect the SFVQ learned codebook to capture a universal morphology of the latent space. As the first experiment, we apply the SFVQ on the intermediate latent space ($\mathcal{W}$) of StyleGAN2 (Karras et al., 2020) pre-trained on the CIFAR10 dataset. During training, the number of extracted latent vectors is unbiased for all CIFAR10 classes. In Fig. 7(a), we plot the generated images corresponding to 6 bit SFVQ codebook ($N = 64$), i.e., each image corresponds to a codebook vector. At first glance, we observe a clear arrangement with respect to the image class, where images from the same category are organized into groups. Also, apart from the *horse* class, all animal types and industrial vehicles are located next to each other. Furthermore, there are some visible similarities for subsequent codewords within a class, such as similar objects' rotation, scale, color, and background.

We also see these clear interpretations consistently when training SFVQ with different bitrates (see Appendix A.8). Furthermore, when increasing the SFVQ bitrate by one (doubling the codebook size), the number of specified codewords to each CIFAR10 class will be approximately doubled, and as a result, the proportion of different classes from the codebook remains unchanged. For instance, the *horse* class is always the dominant class of data in the StyleGAN2 latent space by occupying about 25% of the codewords (see Appendix A.8). The reason is that the latent vectors of the *horse* class have the highest diversity compared to other classes (see Appendix A.1). As a result, when training SFVQ with $N = 4$ initial codewords, and then expanding the codebook from $N = 4$ to a larger codebook, a big portion of codewords tend to model the latent vectors of the *horse* class to minimize the MSE loss (Eq. (2)). However, when initializing the SFVQ with $N = \{8, 16\}$ codewords, the *horse* class would not take up to 25% of the learned SFVQ codebook vectors (see Appendix A.1).

To inspect the learned SFVQ from another viewpoint, we plotted the heatmap of Euclidean distances between all SFVQ codebook vectors in Fig. 7(b). Again, we observe a clear separation between different classes, as each dark box shows a data class. It is essential to note that the SFVQ captures this class separation property due to its inherent orderliness and in a completely unsupervised manner. Additionally, we observe a larger dark box shared between the *cat* and *dog* classes, as they are the most similar classes and reside close to each other in the latent space.

In the second experiment, we applied a 5 bit SFVQ ($N = 32$) on the $\mathcal{W}$ space of the pre-trained StyleGAN2 on the FFHQ dataset. Images corresponding to the SFVQ codebook are represented in Fig. 8(a). We observe similarities among neighboring codebook vectors, such as baby-aged faces for indices 6-7, hat accessory for indices 13-16, eyeglasses for indices 18-19, rotation from right to left from index 17 to 20, and rotation from left to right from index 27 to 31. Based on our investigations, the StyleGAN2's $\mathcal{W}$ space for FFHQ, AFHQ, and LSUN Cars are much denser and entangled than CIFAR10 because they are trained on not very diverse data like CIFAR10. That is why the learned SFVQ curve shown in Fig. 8(a) does not show a perfect distinctive universal interpretation. We provided a similar figure for a 6 bit SFVQ for the AFHQ dataset in Appendix A.2.

As the third experiment, we examine a 2 bit SFVQ ($N = 4$) applied to the $\mathcal{W}$ space of StyleGAN2, pre-trained on the FFHQ dataset, and display the generated images in Fig. 8(b). We observe a clear separation between females and males, with only two unique identities, each representing the average face for females and males. From this SFVQ curve, we can infer some more interesting properties. We hypothesize that each SFVQ line corresponds to an interpretable direction shown in Fig. 8(c). Direction I (the direction from codebook vectors 1 to 2) is for changing rotation to the right, direction II refers to the gender change, and direction III is for changing rotation to the left. We also compute the angles between these directions in degrees, which somehow confirms our hypothesis. Direction II is almost orthogonal to the two other directions, and directions I and III are approximately inverse.

## 5.2 StyleGAN2: interpretable directions

As discussed in Sec. 3.2.3, SFVQ lines that connect the subsequent codewords can refer to meaningful interpretable directions. Hence, we apply SFVQ curves (from 2 to 12 bit) to the $\mathcal{W}$ space of StyleGAN2, pre-trained on the FFHQ, AFHQ, and LSUN Cars datasets, and observe the generated images corresponding to the SFVQ curves. By observation, we spot some useful interpretable directions, shown in Fig. 9. Columns (a) and (b) represent the discovered direction from two SFVQ subsequent codebook vectors, column (c) is the test vector in the latent space to which we apply the direction, and column (d) is the final result after applying the direction. Similar to the GANSpace naming convention, the term $\mathrm{W}i$-$\mathrm{W}j$ means we only manipulate the style blocks within the range $[i\text{-}j]$. Note that we take the directions only from SFVQ's subsequent codebook vectors, but not from two necessarily similar, though far apart, codebook vectors. Otherwise, one can accidentally find directions by taking two codebook vectors from an ordinary VQ that might lead to a meaningful direction. To show the practicality of the directions better, we applied them only on one identical test image (except for the *Beard* and *Bald* directions, which are specified to males).

One significant advantage of our proposed method over other approaches is that it maintains the identity of the test image (column (c)) to a great extent when applying the interpretable directions. Another advantage is that we could find some new and unique directions that were not found in previous methods, such as *Hat*, *Beard* for FFHQ, *Age*, *Bicolor* for AFHQ, and *Classic* for LSUN Cars. These unique directions are not limited only to these, as users can find other directions by their own observations. More importantly, our approach detects an inclusive set of directions, whereas other methods in the literature can only find a portion of them. It is important to note that the directions for the AFHQ dataset are class-agnostic (see Appendix A.3), i.e., the direction for one animal works for other animal species because in Fig. 9 we find the directions from *Wolf* and *Cat* classes, but we apply them to a *Dog* class. However, some directions do not necessarily work for all animal species in the AFHQ because the transformations are restricted by the dataset bias of individual animal classes (Jahanian et al., 2019) (see Appendix A.3). Another interesting observation is how the *Hat* direction (discovered for males) works logically but differently for females.

## 5.3 BigGAN: interpretable directions

BigGAN (Brock et al., 2018) samples a random vector $z$ from a normal prior distribution $p(z)$ and maps it to an image. Since BigGAN's intermediate layers also take the random vector $z$ as input (i.e., *skip-z connections*), the vector $z$ has the most significant effect on the generated output image. Hence, we should

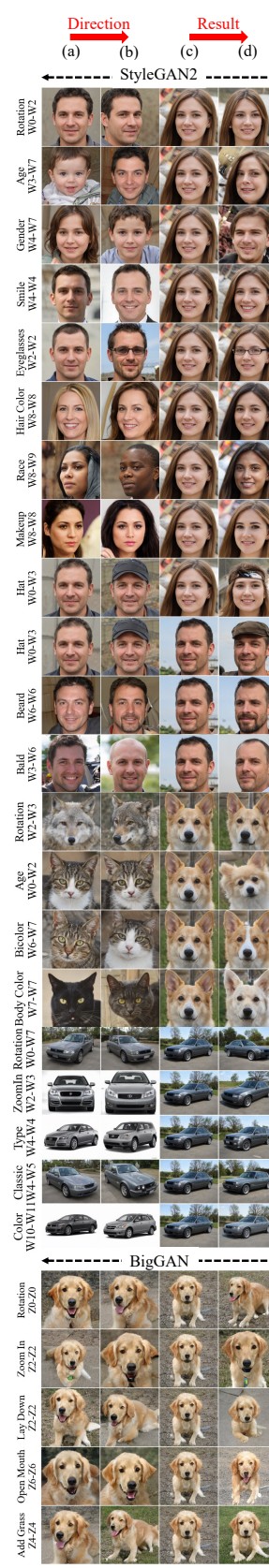

Figure 9: SFVQ interpretable directions.

find the semantic directions in $p(z)$ space. However, as $p(z)$ is an isotropic distribution, it is difficult to find useful directions from it (Härkönen et al., 2020). Therefore, similar to GANSpace, we first train the SFVQ on the first linear layer ($\mathcal{L}$) of BigGAN to search for interpretable directions within this space, and afterward, we transfer these directions back to $p(z)$ space. To this end, we sample $10^6$ random vectors from $p(z)$ and generate their corresponding vectors in $\mathcal{L}$ space. We map these vectors (in $\mathcal{L}$) to the learned SFVQ codebook vectors such that each sample will be mapped to its closest codebook vector using Euclidean distance. Therefore, for each codebook vector in $\mathcal{L}$, we have its corresponding samples in $p(z)$. Finally, for each codebook vector in $\mathcal{L}$, we compute its corresponding codebook vector in $p(z)$ by taking the mean of the vectors in $p(z)$ which get mapped to this SFVQ codebook vector. We obtain the corresponding SFVQ curve in $p(z)$ by doing this operation for all codebook vectors. Now, we use this computed SFVQ codebook (in $p(z)$) to interpret the latent space of BigGAN. Note that to compute the SFVQ curve for BigGAN, we select a class label and keep it fixed.

We computed the SFVQ curve over different bitrates (from 2 to 12 bit) in the $p(z)$ space of BigGAN for *golden retriever* class and discovered some interpretable directions, which are shown in Fig. 9. Columns (a) and (b) represent the discovered direction from two SFVQ subsequent codebook vectors, column (c) is the test vector in the $p(z)$ to which we apply the direction, and column (d) is the final result after applying the direction. Similar to the GANSpace naming convention, the term Z*i*-Z*j* means we only manipulate the *skip-z connections* within the range [*i-j*]. Apart from basic geometrical directions (*Rotation* and *Zoom In*), we discovered some more specific directions such as *Lay Down* and *Open Mouth* as found in Yüksel et al. (2021), and *Add Grass* as found in Härkönen et al. (2020). Note that the discovered directions by SFVQ for *golden retriever* class are class-agnostic, i.e., they also work for other classes (see Appendix A.4).

### 5.4 Qualitative comparison

We compared our interpretable directions with GANSpace (Härkönen et al., 2020), LatentCLR (Yüksel et al., 2021), and SeFa (Shen & Zhou, 2021) qualitatively and quantitatively. The reason for choosing these methods is that their interpretable directions for StyleGAN2-FFHQ were readily available in their GitHub repositories. Hence, we skipped other methods that were not trained on StyleGAN2-FFHQ or did not share their directions. We focus on StyleGAN2-FFHQ for comparisons, as we planned to use the pre-trained networks of Zhang et al. (2017); Karkkainen & Joo (2021); Jiang et al. (2021); Doosti et al. (2020); Deng et al. (2019) for face attribute rating in our quantitative comparisons. For SeFa, there were no annotations for the discovered directions of StyleGAN2-FFHQ. Hence, we used their interactive tool to examine their first $K = 25$ semantics and identify their interpretable directions. In SeFa interactive tool, there were only three

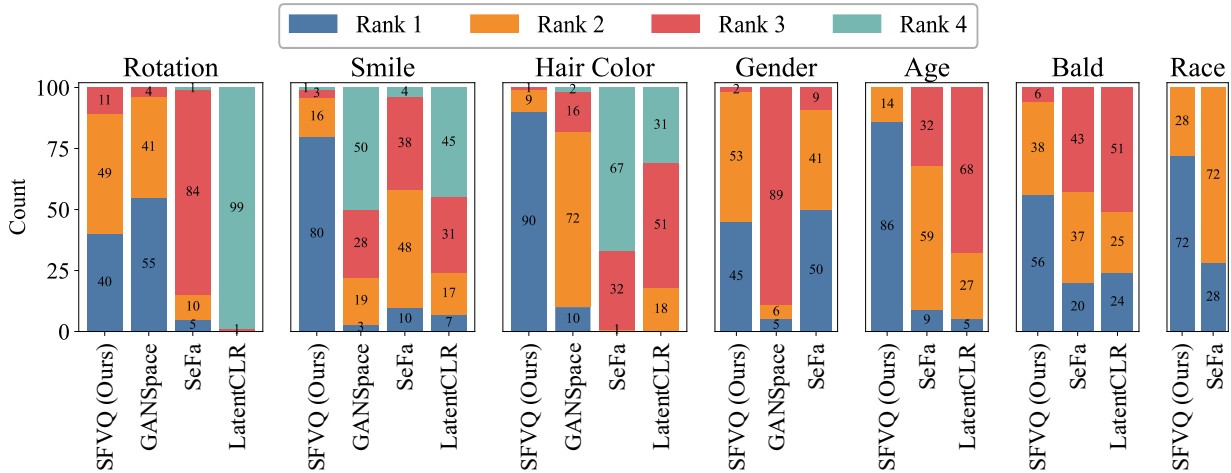

Figure 10: Subjective test results by asking 20 human subjects to rate the interpretable directions of our proposed method, GANSpace, SeFa, and LatentCLR from best (Rank 1) to worst (Rank 4). These seven directions are assessed using 100 random latent vectors, with 5 latent vectors assigned to each human subject.

possible options for the layer-wise edits ($W0$-$W1$, $W2$-$W5$, $W6$-$W13$), but to have a fair comparison, we further searched precisely and found the best layer-wise edits for each direction. We provided the information of SeFa interpretable directions in Appendix A.6.

Fig. 11 shows the qualitative comparison. We provided a similar comparison over 50 different random vectors in the Supplementary Material (shown as $\mathcal{SM}$ in the rest of the paper). To have a fair comparison, we use the same amount of shift ($\sigma$) toward each direction because, as mentioned in Tzelepis et al. (2021), it is the advantage of a direction if it reaches the desired change in the attribute within a shorter path. The image in the red square is the initial test image to which we apply the changes. For the *Smile* direction, our method effectively opens and closes the smile in both positive and negative paths while maintaining the identity and age attributes with minimal changes. Whereas GANSpace is highly entangled with the age attribute and SeFa changes the identity. In *Hair Color* direction, our method keeps the identity better than the others. However, GANSpace and SeFa alter the face highlights, LatentCLR is highly entangled with gender, and SeFa is highly entangled with age. For *Age* direction, our method covers a wider range of ages than LatentCLR, while LatentCLR and SeFa are highly entangled with gender, and SeFa alters the identity too much. For *Gender* direction, GANSpace and SeFa are entangled with age, and our method remains in the valid range of generations better than GANSpace. In *Bald* direction, our method keeps the identity better than SeFa, while SeFa adds beard to the face, and our method renders a better baldness than Latent-CLR. To see a more comprehensive qualitative comparison, we encourage the readers to make subjective comparisons with different random vectors using our GitHub repository or to inspect subjective comparisons over 50 random vectors in $\mathcal{SM}$.

We conducted a subjective test to compare the interpretable directions of various methods over 100 different random vectors. For each random vector and each direction, we generated the edited images with the steps of $\{-3\sigma, -2\sigma, -1\sigma, 0\sigma, 1\sigma, 2\sigma, 3\sigma\}$ for all methods, where $\sigma = 2.67$. We delivered the generated images of 5 different random vectors to each of 20 human subjects, and asked them to rate the interpretable directions by answering this question: "*Sort the methods that apply the desired change on the test image convincingly, and simultaneously keep the other attributes of the test image (especially the identity) fixed*". In the case of $M$ different methods, the subject would rate them by assigning a number from $\{1, \ldots, M\}$, where the best is ranked 1 and the worst is ranked $M$. Fig. 10 shows the results of the subjective test. Based on the results, our method clearly outperforms the other methods for *Smile*, *Hair Color*, *Age*, *Bald*, and *Race* directions. For *Rotation* and *Gender* directions, our method performs comparably to GANSpace and SeFa, respectively.

## 5.5 Quantitative comparison

For quantitative comparison, we adopted the evaluation criteria and pre-trained networks used in Tzelepis et al. (2021) and Aoshima & Matsubara (2023) to rate an image's attributes. We use Zhang et al. (2017) to spot the face bounding box, FairFace (Karkkainen & Joo, 2021) to rate the age, race, and gender attributes, CelebA-HQ (Jiang et al., 2021) to measure the smile attribute, Hopenet (Doosti et al., 2020) to find the face direction (yaw, pitch, roll attributes), and ArcFace (Deng et al., 2019) to evaluate how much the face identity is preserved after shifting along a direction.

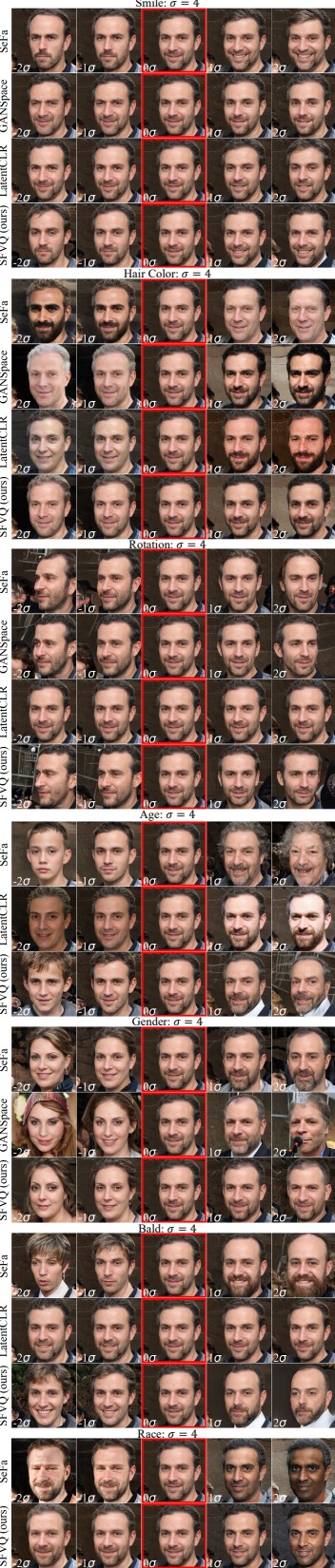

Figure 11: Qualitative Comparison.

We sample $10^3$ vectors from $\mathcal{N}(0,1)$ and generate their corresponding latent vectors in the $\mathcal{W}$ space of StyleGAN2. Following Tzelepis et al. (2021), to assess a discovered direction for each latent vector, we create a sequence of images by shifting the latent vector for 20 steps in both positive and negative paths along that direction. Therefore, each sequence contains 41 images, with the original intact image positioned in the middle. Then, for each image within this sequence, we use the above-mentioned pre-trained networks to measure its attributes. Next, we calculate the correlation between the step indices (from 1 to 41) and each attribute's scores. Thus, for each direction, we obtain a vector of seven correlation values (one per each attribute) which is then L1-normalized, similar to Tzelepis et al. (2021). Table 1 shows the results averaged over $10^3$ latent vectors for our method, GANSpace, LatentCLR and SeFa. Values are shown in *red* if the direction is correlated the most with a wrong attribute.

Table 1 shows that for *Gender* direction, our method works better than the others with a higher correlation to gender attribute, whereas GANSpace and SeFa are correlated with age and race attributes. For the *Age* direction, our method has almost the

Table 1: Quantitative comparison of interpretable directions. Values in each row show the L1-normalized correlation of each direction to the attributes. Best values are shown in *Bold*, and values in *red* mean the direction is correlated the most with a wrong attribute.

| Direction | Method | Gender | Age | Smile | Race | Yaw | Pitch | Roll |
|---|---|---|---|---|---|---|---|---|
| Gender | GANSpace | 0.63 | 0.12 | 0.047 | 0.13 | 0.0093 | 0.05 | 0.0074 |
|  | SeFa | 0.74 | 0.11 | 0.06 | 0.06 | 0.018 | 0.0056 | 0.01 |
|  | SFVQ (Ours) | **0.87** | 0.0027 | 0.037 | 0.039 | 0.011 | 0.031 | 0.0052 |
| Age | LatentCLR | 0.31 | **0.38** | 0.034 | 0.24 | 0.014 | 0.0049 | 0.0067 |
|  | SeFa | *0.48* | 0.25 | 0.081 | 0.14 | 0.0074 | 0.029 | 0.013 |
|  | SFVQ (Ours) | 0.11 | 0.37 | 0.14 | 0.018 | 0.09 | 0.24 | 0.057 |
| Smile | GANSpace | 0.047 | *0.53* | 0.0052 | 0.36 | 0.0005 | 0.057 | 0.0008 |
|  | LatentCLR | 0.15 | 0.15 | 0.17 | 0.1 | 0.052 | *0.32* | 0.056 |
|  | SeFa | 0.26 | 0.12 | **0.46** | 0.023 | 0.0011 | 0.12 | 0.0047 |
|  | SFVQ (Ours) | 0.31 | 0.078 | 0.4 | 0.077 | 0.022 | 0.052 | 0.061 |
| Race | SeFa | 0.066 | 0.17 | 0.34 | 0.37 | 0.0057 | 0.027 | 0.025 |
|  | SFVQ (Ours) | 0.037 | 0.07 | 0.33 | **0.52** | 0.0015 | 0.031 | 0.0011 |
| Rotation | GANSpace | 0.11 | 0.037 | 0.0023 | 0.022 | **0.76** | 0.063 | 0.0032 |
|  | LatentCLR | 0.12 | 0.051 | 0.11 | 0.0027 | 0.67 | 0.032 | 0.01 |
|  | SeFa | 0.3 | 0.024 | 0.01 | 0.0049 | 0.51 | 0.15 | 0.0036 |
|  | SFVQ (Ours) | 0.084 | 0.013 | 0.16 | 0.021 | 0.58 | 0.07 | 0.072 |
| Bald | LatentCLR | 0.25 | 0.083 | 0.17 | 0.16 | 0.032 | 0.23 | 0.073 |
|  | SeFa | 0.24 | 0.32 | 0.053 | 0.31 | 0.024 | 0.013 | 0.043 |
|  | SFVQ (Ours) | 0.47 | 0.14 | 0.12 | 0.019 | 0.059 | 0.16 | 0.027 |
| HairColor | GANSpace | 0.006 | 0.084 | 0.47 | 0.4 | 0.012 | 0.014 | 0.0083 |
|  | LatentCLR | 0.33 | 0.099 | 0.27 | 0.22 | 0.0009 | 0.03 | 0.046 |
|  | SeFa | 0.027 | 0.4 | 0.43 | 0.08 | 0.02 | 0.013 | 0.027 |
|  | SFVQ (Ours) | 0.11 | 0.015 | 0.47 | 0.38 | 0.0022 | 0.012 | 0.019 |

same correlation with the age attribute as LatentCLR, but is higher than SeFa. However, LatentCLR and SeFa remarkably alter the gender and race attributes, which is undesirable. In contrast, our method modifies the smile and pitch attributes, which are visually more acceptable (see Fig. 11). For *Smile* direction, SeFa renders a higher correlation to the smile attribute than others. However, the smile direction of GANSpace and LatentCLR methods are improperly correlated the most with age and pitch attributes, respectively. In *Rotation* direction, similar to other methods, our method is mainly correlated with the yaw attribute but with less correlation than GANSpace and LatentCLR. At the same time, it changes other face rotations' attributes (i.e., pitch and roll) more than they do. Our *Rotation* direction causes fewer changes in gender attributes compared to others, specifically SeFa. Note that in Table 1, if a method is not listed for a direction, it means that direction does not exist for the method.

We also compare our method with others on how they preserve identity when shifting latent vectors for various shift values in different directions. Table 2 provides the identity scores (averaged over $10^3$ latent vectors) that range from 0 to 1, such that a higher value means a higher similarity to the original test image in terms of identity. We observe that our method maintains the identity better than others by a significant margin for *Smile*, *Race*, and *Hair Color* directions. For the *Gender* direction, our method outperforms GANSpace and is comparable to SeFa. In *Age* direction, our method performs comparably to LatentCLR and SeFa. Based on qualitative comparisons (Fig. 11), since LatentCLR applies minimal changes to the face compared to others, it gives higher identity scores for *Rotation* and *Bald* directions. Ignoring the LatentCLR scores, our method performs comparably to GANSpace and SeFa in *Rotation* direction, and better than SeFa for *Bald* direction. Furthermore, we computed the *commutativity error* (defined in Aoshima & Matsubara (2023)) for our method, GANSpace, LatentCLR, and SeFa over all directions. As expected, all four methods are *commutative* because they all apply linear transformations on the latent vectors. Ultimately, we believe that the most effective way to compare the interpretable directions between different methods remains subjective comparisons. Therefore, to better assess the efficiency of our discovered directions compared to other methods, we encourage readers to make subjective comparisons with different random vectors using our GitHub demo directory or to inspect subjective comparisons over 50 random vectors in $\mathcal{SM}$.

## 5.6 Ablation study on SFVQ bitrate

We conducted an ablation study on the effect of different SFVQ bitrates (ranging from 2 to 12 bit) on the interpretations of pre-trained StyleGAN2 models on the FFHQ, AFHQ, and LSUN Cars datasets. Regarding *universal interpretation*, for all bitrates, we observe the inherent structure in SFVQ's subsequent codebook vectors, which share similar generative factors, such as rotation, background, and accessories, for FFHQ. When increasing the bitrate, we see more diversity in the images (e.g., more identities for FFHQ) because we model the latent space with more clusters (or codebook vectors). We provide images corresponding to SFVQ codebooks from 2 to 8 bit in Appendix A.7.

Regarding the *interpretable directions*, a higher SFVQ bitrate allows the curve to get more turned and twisted in the latent space, increasing the chance of spotting more detailed or intricate directions. Based on our investigations, the directions that alter images more structurally can be found from lower bitrates and vice versa. For example, for StyleGAN2-FFHQ, we found *rotation*, *gender* and *age* directions from 2, 5, and 6 bit SFVQ, respectively. On the other hand, we detected the directions that cause a partial change on the face, such as *smile*, *hair color*, *makeup*, *race*, and *bald* from 12 bit SFVQ.

Table 2: Identity preservation scores of interpretable directions for our proposed method, GANSpace, LatentCLR, and SeFa. Values in a row show the identity scores for different shifts ($\sigma$). Best values are shown in *Bold*.

| Direction | Method | 1-4$\sigma$ | 5-8$\sigma$ | 9-12$\sigma$ | 13-16$\sigma$ | 17-20$\sigma$ |
|---|---|---|---|---|---|---|
| Gender | GANSpace | 0.85 | 0.58 | 0.39 | 0.22 | 0.078 |
| | SeFa | **0.94** | **0.78** | **0.63** | **0.51** | **0.4** |
| | SFVQ (Ours) | 0.93 | 0.76 | 0.61 | 0.47 | 0.35 |
| Age | LatentCLR | 0.93 | 0.73 | 0.53 | 0.39 | **0.29** |
| | SeFa | **0.95** | **0.78** | **0.59** | **0.42** | **0.29** |
| | SFVQ (Ours) | 0.94 | 0.76 | 0.56 | 0.39 | 0.25 |
| Smile | GANSpace | 0.94 | 0.76 | 0.57 | 0.41 | 0.29 |
| | LatentCLR | 0.95 | 0.8 | 0.63 | 0.47 | 0.32 |
| | SeFa | 0.93 | 0.78 | 0.62 | 0.47 | 0.35 |
| | SFVQ (Ours) | **0.96** | **0.85** | **0.72** | **0.59** | **0.48** |
| Race | SeFa | 0.93 | 0.7 | 0.48 | 0.33 | 0.22 |
| | SFVQ (Ours) | **0.98** | **0.88** | **0.74** | **0.6** | **0.48** |
| Rotation | GANSpace | 0.93 | 0.77 | 0.62 | 0.51 | 0.42 |
| | LatentCLR | **0.98** | **0.92** | **0.85** | **0.79** | **0.75** |
| | SeFa | 0.93 | 0.77 | 0.62 | 0.5 | 0.4 |
| | SFVQ (Ours) | 0.93 | 0.76 | 0.61 | 0.49 | 0.4 |
| Bald | LatentCLR | **0.96** | **0.86** | **0.73** | **0.61** | **0.51** |
| | SeFa | 0.95 | 0.8 | 0.64 | 0.49 | 0.36 |
| | SFVQ (Ours) | **0.96** | 0.84 | 0.7 | 0.55 | 0.41 |
| HairColor | GANSpace | 0.98 | 0.88 | 0.75 | 0.62 | 0.51 |
| | LatentCLR | 0.96 | 0.81 | 0.63 | 0.47 | 0.36 |
| | SeFa | 0.97 | 0.86 | 0.71 | 0.56 | 0.44 |
| | SFVQ (Ours) | **0.99** | **0.97** | **0.94** | **0.89** | **0.83** |

## 5.7 Joint interpretable directions

By observing images of the learned SFVQ curve (Fig. 8(a)) to find interpretable directions, we can also discover joint interpretable directions from subsequent codebook vectors that differ in multiple attributes. By *joint*, we mean to change, for example, *rotation* and *gender* attributes simultaneously. Joint directions are the directions in which multiple attributes are entangled. Supervised methods cannot find joint directions because they use pre-trained networks or labeled data with respect to only one attribute. Furthermore, finding joint directions will be laborious for the unsupervised methods of Härkö-nen et al. (2020); Shen & Zhou (2021); Voynov & Babenko (2020); Yüksel et al. (2021); Tzelepis et al. (2021); Aoshima & Matsubara (2023) because 1) their training strategy is not designed for this task, 2) they have to blindly search over all $K$ detected directions and hope to find the direction to change their desirable joint attributes. However, in our method, the

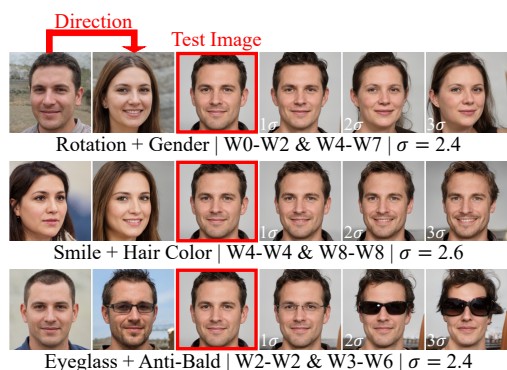

Rotation + Gender | W0-W2 & W4-W7 | $\sigma = 2.4$

Smile + Hair Color | W4-W4 & W8-W8 | $\sigma = 2.6$

Eyeglass + Anti-Bald | W2-W2 & W3-W6 | $\sigma = 2.4$

Figure 12: Some examples of SFVQ joint interpretable directions.

prior knowledge of potential directions obtained by observing the SFVQ curve helps to quickly identify the desirable joint directions. Fig. 12 shows some joint directions found by our proposed method. Note that the joint directions are not limited to these, as users can discover their desired directions by their own inspections.

## 5.8 Controllable data augmentation

According to the training objective of SFVQ, to map input vectors onto the line connecting subsequent codebook vectors, SFVQ has the property that its lines are mainly located within the distribution's space. This property is desirable for controllable data augmentation because we have many meaningful points (located on the SFVQ curve) available to generate valid images. By looking at images corresponding to

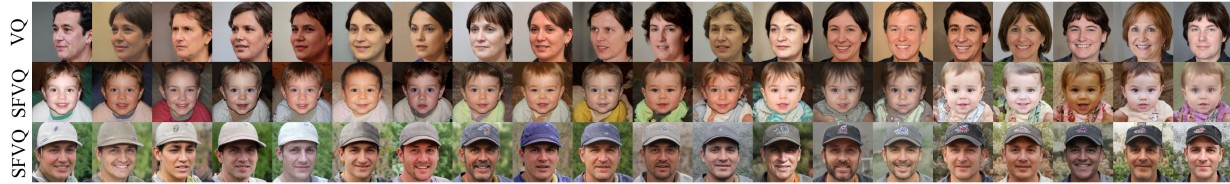

Figure 13: Generated images from 20 equally-spaced points on the line connecting two neighboring codebook vectors of VQ and two subsequent codebook vectors of SFVQ trained on the $\mathcal{W}$ space of StyleGAN2 pre-trained on FFHQ dataset.

the SFVQ curve in Fig. 8(a), we have an idea of the possible generations from each part of the curve. For instance, to generate baby-aged faces, we select 20 equally-spaced points on the line connecting codebook vectors of indices 6 to 7 in Fig. 8(a), and we plot the generations corresponding to these 20 points in the middle row of Fig. 13. Similarly, we take 20 equally-spaced points on the line connecting two subsequent codebook vectors of indices 15 and 16 in Fig. 8(a) and generate their corresponding images in the bottom row of Fig. 13. We observe that all 20 generations contain the *hat* accessory for a male person.

We also take the line connecting two neighboring codebook vectors (under Euclidean distance) of a 5 bit VQ and plot similar generations in the top row of Fig. 13. To obtain a more diverse representation of generations, for all generations in Fig. 13, we added normal noise ($\mathcal{N}(0, 0.3)$) to the selected points. As expected, all generations of SFVQ consistently follow the properties of their corner points, such that they are all faces of babies or males wearing hats. However, the generations for two neighboring codebook vectors of VQ do not follow any specific rule as we observe changes in gender, age, and race among them. Thus, here, by *controllable*, we mean that the users have control over what type of images with specific characteristics they intend to generate.

## 6 Conclusions

Generative adversarial networks (GANs) are well-known image synthesis models widely used to generate high-quality images. However, there is still insufficient control over generations in GANs because their latent spaces act as a black box, making them hard to interpret. In this paper, we use the unsupervised space-filling vector quantizer (SFVQ) technique to obtain a universal interpretation of the latent spaces of GANs and to find their interpretable directions. Our experiments demonstrate that the SFVQ can capture the underlying morphological structure of the latent space and discover more effective and consistent interpretable directions compared to GANSpace, LatentCLR, and SeFa methods. SFVQ provides the user with proper control over generating and manipulating images, and reduces the effort required to find the desired direction of a change. SFVQ is a generic tool for modeling distributions that is neither restricted to any specific neural network architecture nor any data type (e.g., image, video, speech, etc.).

## Acknowledgments

We gratefully acknowledge the participants for their involvement in the subjective evaluation tests. We also acknowledge the computational resources provided by the Aalto Science-IT project.

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

# A    Appendices

## A.1    StyleGAN2: universal interpretation of the CIFAR10 dataset

As discussed in Sec. 5.1, when we initialize the SFVQ with $N = 4$ initial codewords and train it on the $\mathcal{W}$ space of StyleGAN2 pre-trained on CIFAR10 dataset, about 25% of the learned codewords would model the *horse* class (see Fig. 7(a)). To find out the reason, for each CIFAR10 class, we extracted 10 k latent vectors, and computed three metrics: the mean and variance of Euclidean distances among all latent vectors, and the sum of the eigenvalues of the latent vectors. Fig. 14 shows the computations of these three metrics for all CIFAR10 different classes. According to the figure, the *horse* class takes the widest area of the latent space while its latent vectors have the highest diversity compared to other classes. However, the computed metrics for the *horse* class do not show a significant difference from other classes, and thus it does not conform to the fact that it accounts for 25% of the learned codewords. Therefore, we train SFVQ with higher numbers of initial codewords ($N = \{8, 16\}$), and we plot the generated images corresponding to the learned SFVQ codebook in Fig. 15 and Fig. 16 where SFVQ was initialized with $N = 8$ and $N = 16$ codewords, respectively. In these cases, we observe a more normalized proportion of CIFAR10 classes occupying the codebook, which better aligns with the computed metrics in Fig. 14.

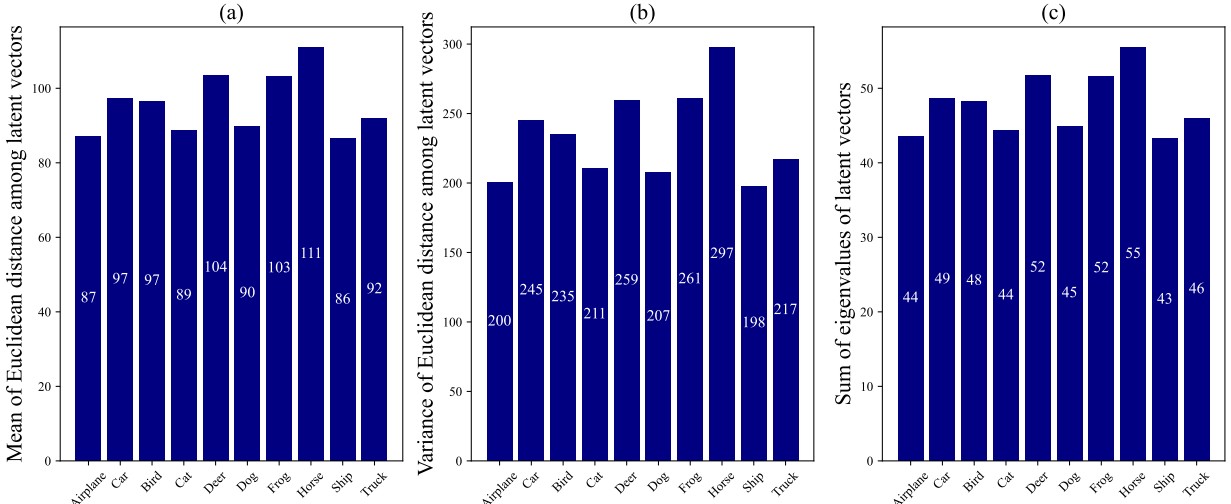

Figure 14: (a) Mean and (b) variance of Euclidean distances among 10 k latent vectors of each CIFAR10 class, and (c) sum of the eigenvalues of the latent vectors in $\mathcal{W}$ space of pre-trained StyleGAN2.

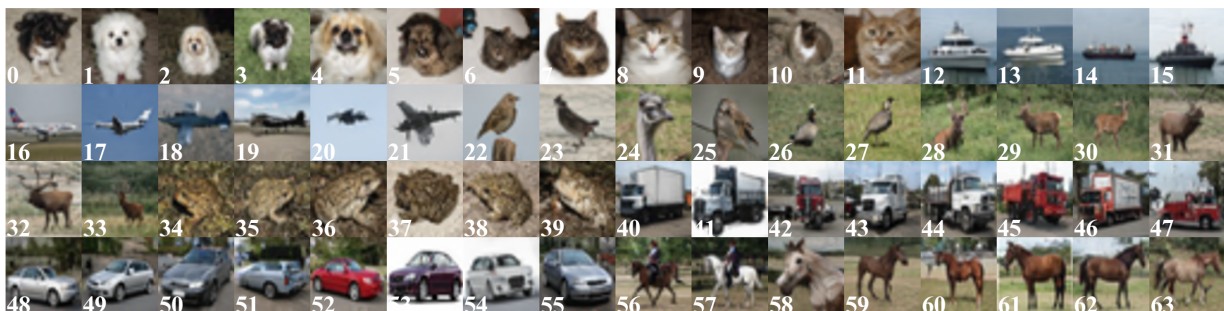

Figure 15: Generated images from the codebook of a 6 bit SFVQ ($N = 64$) trained on $\mathcal{W}$ space of StyleGAN2 pre-trained on the CIFAR10 dataset, when SFVQ is initialized by $N = 8$ codewords.

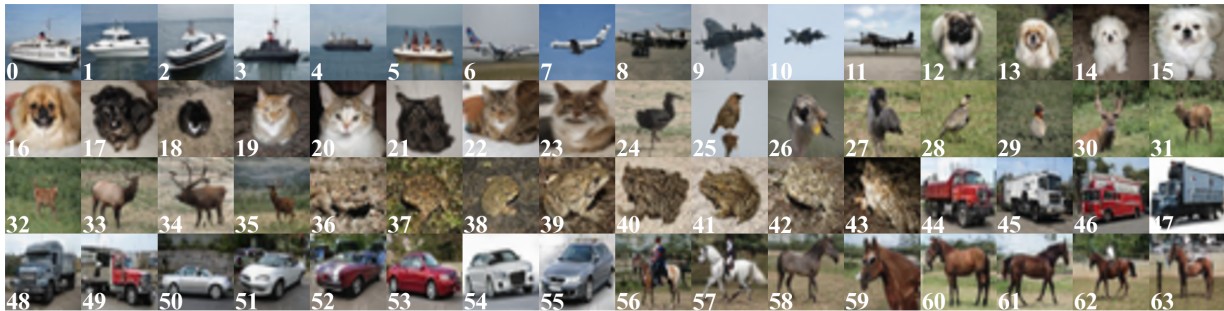

Figure 16: Generated images from the codebook of a 6 bit SFVQ ($N = 64$) trained on $\mathcal{W}$ space of StyleGAN2 pre-trained on the CIFAR10 dataset, when SFVQ is initialized by $N = 16$ codewords.

## A.2 StyleGAN2: universal interpretation of the AFHQ dataset

Similar to what is discussed in Sec. 5.1 of the paper, we apply the SFVQ to capture a universal morphology of the latent space, and we expect that subsequent codebook vectors in SFVQ refer to similar images. Hence, we applied a 6 bit SFVQ on the $\mathcal{W}$ space of the StyleGAN2 model, which was pre-trained on the AFHQ dataset. Images corresponding to the SFVQ's codebook vectors are represented in Fig. 17. We can observe that similar animal species are typically located adjacent to one another. In addition, there are some other similarities among neighboring codebook vectors, such as change in rotation (from right to left) when moving from index 0 to index 10, change in rotation (from left to right) when moving from index 26 to index 34, light-colored animals for indices 22-25, bi-colored animals for indices 26-29, and baby-aged cats for indices 61-62.

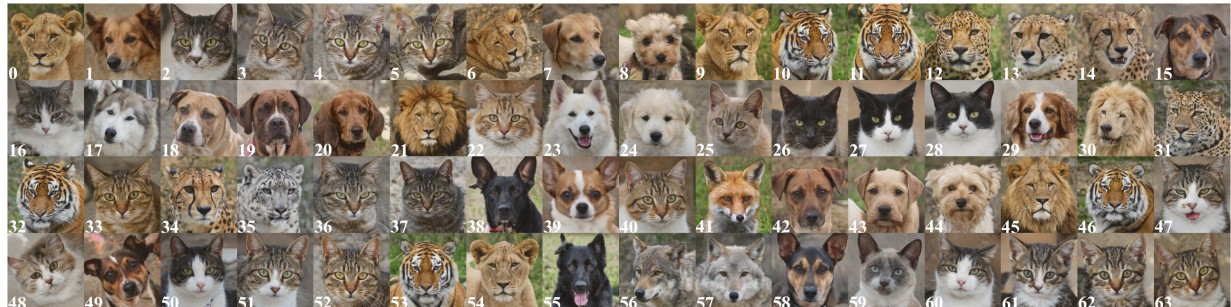

Figure 17: Generated images from the codebook of a 6 bit SFVQ ($N = 64$) trained on $\mathcal{W}$ space of StyleGAN2 pre-trained on the AFHQ dataset.

### A.3 Class-agnostic directions for StyleGAN2 pre-trained on the AFHQ dataset

According to what is discussed in Sec. 5.2 of the paper, in this section, we aim to test whether and how the discovered direction of *Bicolor* (in Fig. 9 of the paper) is class-agnostic across different AFHQ animal classes. To this end, we applied this direction to all existing animal species in the AFHQ dataset and represented the results in Fig. 18. We observe that this direction works well for the *Cat* and *Dog* classes because there is sufficient data (i.e., cats and dogs with bicolored faces) within the AFHQ dataset. Therefore, the learned latent space supports this transformation. In addition, this transformation more or less works for *Wolf* class, since *Wolf* looks like *Siberian husky* (which exists in AFHQ dataset), and this transformation leads the *Wolf* class to become similar to a *Siberian husky*. However, the *Bicolor* direction does not work for other animal classes of *Fox*, *Leopard*, *Cheetah*, *Tiger*, and *Lion*. The reason is that the learned latent space is constrained by the dataset bias of individual classes (Jahanian et al., 2019). In other words, the learned latent space does not support this transformation for them, as there are no images with a bicolored face from these animal classes within the AFHQ dataset. The $\sigma$ value determines the magnitude of the step we take toward the *Bicolor* direction. To make sure whether this direction works for these five animal classes, we used a larger $\sigma$ value (bigger steps) for them. We observe that even with larger steps, not only is there no meaningful transformation effect in the desired direction, but also, in the very last step ($3\sigma$), the images become unrealistic due to the presence of artifacts.

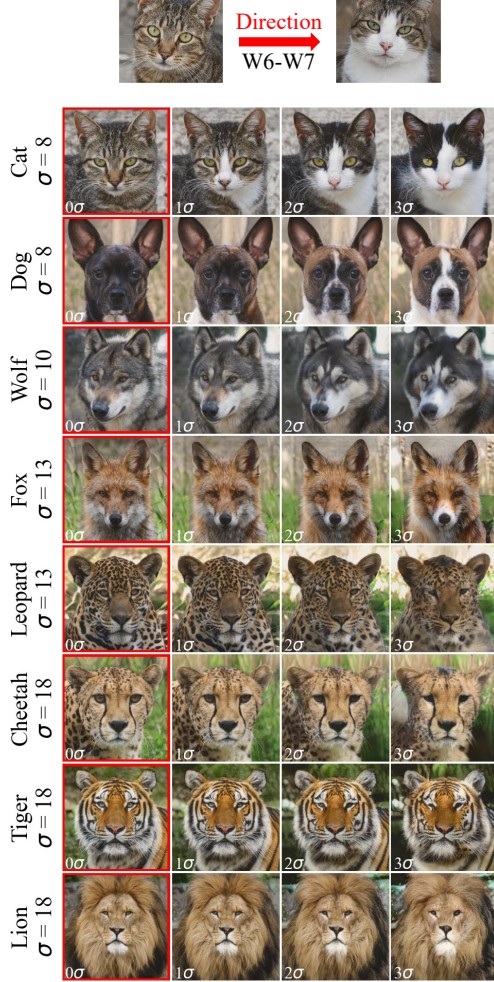

Figure 18: Applying *Bicolor* direction to different animal species of AFHQ dataset.

### A.4 Class-agnostic directions for BigGAN pre-trained on the ImageNet dataset

As discussed in Sec. 5.3 of the paper, we find out that the discovered directions by SFVQ (in $p(z)$ space of BigGAN) for the *golden retriever* class are class-agnostic. It means that the detected directions also work when applied to other data classes within the ImageNet dataset. To confirm this, we applied all five directions found for the *golden retriever* (in Fig. 9 of the paper) on the *husky* class, and we illustrated the results in Fig. 19. The image in the middle column (in red square) is the initial test image to which we apply the directions, such that we step along both sides of a direction. According to the figure, all five directions are valid for the *husky* class, resulting in meaningful and expected transformations.

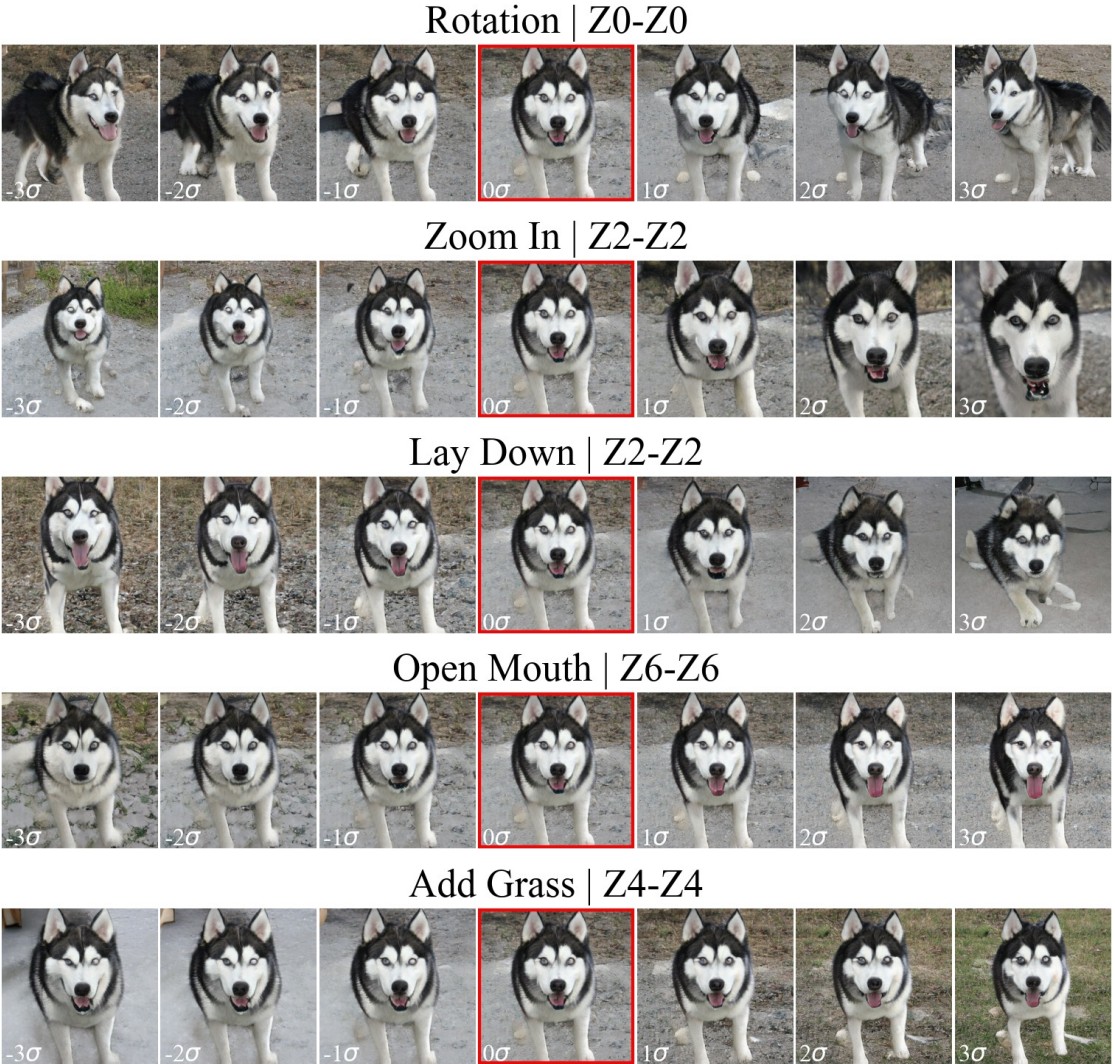

Figure 19: Class-agnostic directions; applying five SFVQ's discovered directions for the *golden retriever* class on the *husky* class using BigGAN pre-trained on the ImageNet dataset.

### A.5 Subsidiary study: traveling salesman problem

Space-filling vector quantization (SFVQ) has some parallels with the classic *traveling salesman* problem (TSP) (Flood, 1956) in bringing order to a set of codebook vectors. One could ask whether we can achieve a better codebook arrangement than SFVQ by applying an ordinary vector quantization (VQ) and afterward, use one of the *traveling salesman* solutions to reorganize VQ codebook vectors. The scenario of TSP involves a list of cities (codebook vectors) and the distances between them, with the goal of discovering the shortest possible route to visit each city only once. TSP is an NP-hard problem to solve. We can interpret these cities as the codewords of a VQ codebook. If we learn an 8 bit VQ as usual and intend to rearrange the codebook vectors to achieve the shortest route, then there are 256! possible permutations for rearrangement. This is an astronomically large number ($8.5 \times 10^{506}$). It is thus practically infeasible to do an exhaustive search for all possible permutations in most relatively high-bitrate cases of VQ. Hence, it is recommended to use heuristic TSP solvers that have lower computational complexity, such as nearest neighbor (Johnson & McGeoch, 1997), greedy (Johnson & McGeoch, 1997), and Christofides (Christofides, 1976).

To compare the performance of TSP heuristic solutions with the SFVQ, we examine their ability to model three sparse sample distributions of *circles*, *moons*, and *spiral* in 3D space. We chose the distributions to be sparse because it makes the task more challenging. We trained ordinary VQ and SFVQ with 9 bit (with

identical initialization and hyper-parameter settings). After training the VQ, we rearranged its codebook vectors using the nearest neighbor (NN) and Christofides TSP heuristic solvers. Fig. 20 demonstrates the results such that the order in the space-filling line is shown with color coding (light to dark color = first to last codebook vector) for both methods of VQ+TSP and SFVQ. Since the training objective of SFVQ is different from VQ, SFVQ locates the codebook vectors such that the line connecting the subsequent codebook vectors mainly desires to fill up the distribution space. As a result, the line ends up landing inside the distribution space. To affirm this fact, compare the upper and lower parts of *spiral* dataset arranged by VQ+Christofides and VQ+NN methods. VQ locates fewer codebook vectors for these two parts of the *spiral* data, and thus we observe a narrow line that does not fill the distribution's space appropriately. Furthermore, we notice more unfavorable jumps (lines outside the distribution or lines breaking the arrangement) for VQ+TSP methods than the SFVQ due to their improper codebook arrangement. Therefore, we generally observe that the SFVQ achieves a much better codebook arrangement than VQ+TSP for all three distributions.

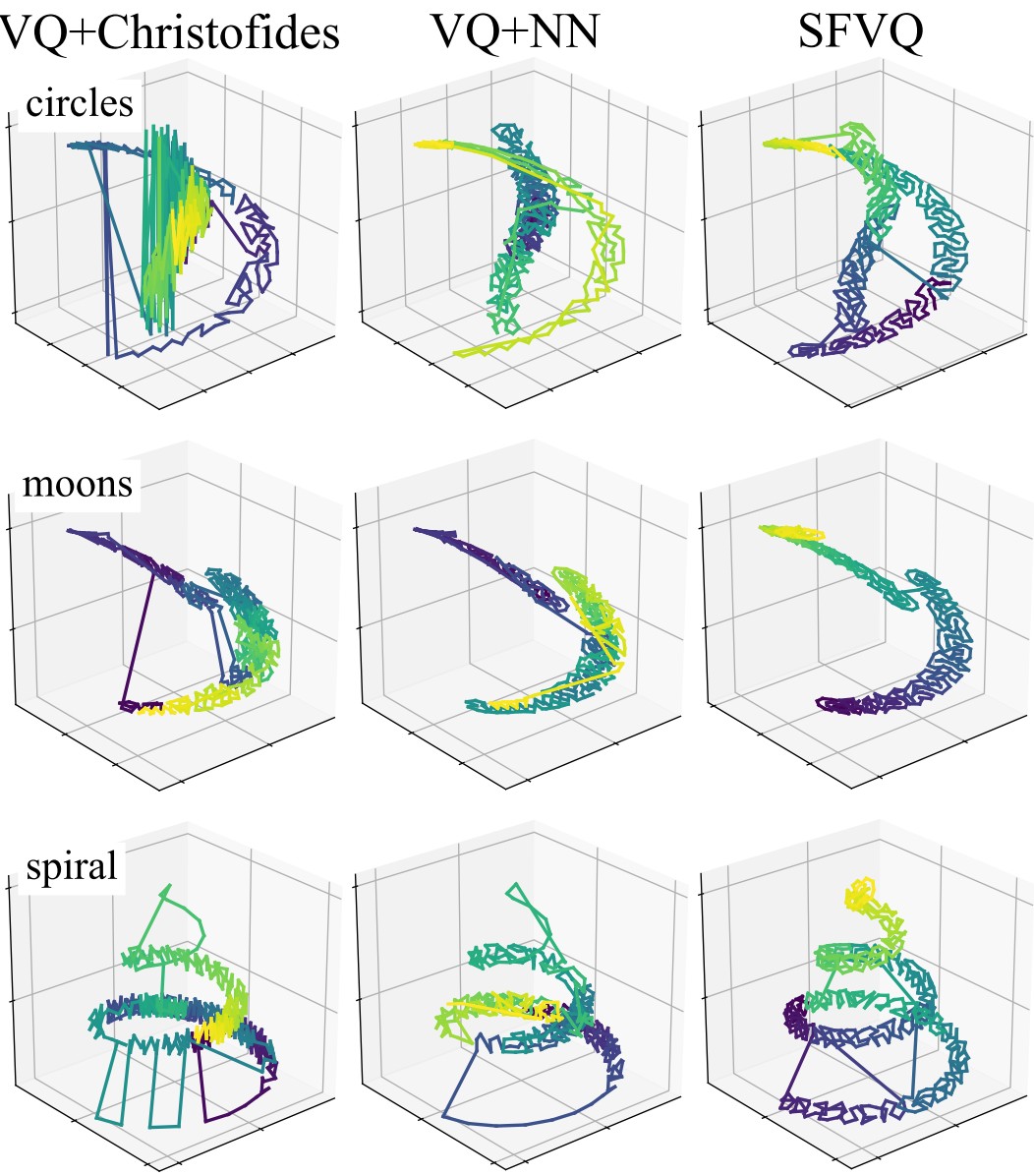

Figure 20: Comparison of the codebook arrangement property of SFVQ with ordinary VQ which is post-processed by *traveling salesman* heuristic solvers over three sparse distributions.

### A.6 SeFa interpretable directions

There were no annotations for SeFa (Shen & Zhou, 2021) interpretable directions for StyleGAN2, which was pre-trained on the FFHQ dataset. Therefore, we used the interactive tool provided in SeFa's GitHub repository and examined the first $K = 25$ semantics of the StyleGAN2-FFHQ model. In the interactive tool, there were only three options available ($W0$-$W1$, $W2$-$W5$, $W6$-$W13$) to do the layer-wise edits and manipulate the latent vector only for those layers. Hence, to obtain more precise interpretable directions, we further searched for the best layer-wise edits for each semantic (or interpretable direction). We provided the details of SeFa's interpretable directions in Table 3.

Table 3: SeFa (Shen & Zhou, 2021) interpretable directions.

| Direction | Semantic Index | StyleGAN2 Layers |
|---|---|---|
| Gender | 2 | [4-6] |
| Rotation | 5 | [0-2] |
| Eyeglasses | 8 | [0-1] |
| Race | 8 | [6-11] |
| Age | 9 | [3-4] |
| Bald | 15 | [4-5] |
| Hair Color | 18 | [7-8] |
| Smile | 21 | [4-5] |
| Beard | 21 | [8-9] |
| Fat | 22 | [2-5] |
| Makeup | 24 | [6-7] |

### A.7 Learned SFVQ codebooks for StyleGAN2 pre-trained on the FFHQ dataset

As mentioned in Sec. 5.6, we provide the learned SFVQ curves trained on the $\mathcal{W}$ space of pre-trained StyleGAN2 on the FFHQ dataset here. Fig. 21 to Fig. 27 demonstrate the generated images from learned SFVQ codebooks with the bitrates from 2 to 8 bit. We also provided similar figures for bitrates of 9 to 12 bit in our GitHub repository. The learned SFVQ codebooks and their corresponding generated images for pre-trained StyleGAN2 on the FFHQ, AFHQ, and LSUN Cars for bitrates ranging from 2 to 12 bit are available in our GitHub repository.

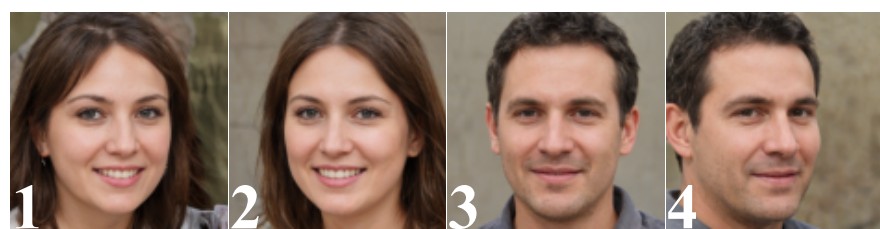

Figure 21: Codebook of a 2 bit SFVQ trained on $\mathcal{W}$ space of StyleGAN2 pre-trained on the FFHQ dataset.

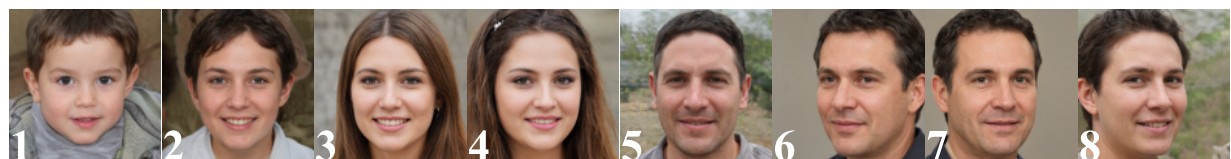

Figure 22: Codebook of a 3 bit SFVQ trained on $\mathcal{W}$ space of StyleGAN2 pre-trained on the FFHQ dataset.

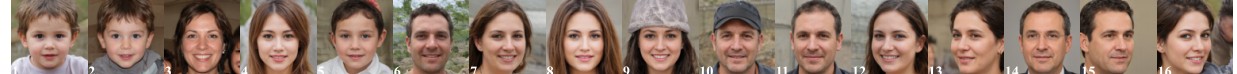

Figure 23: Codebook of a 4 bit SFVQ trained on $\mathcal{W}$ space of StyleGAN2 pre-trained on the FFHQ dataset.

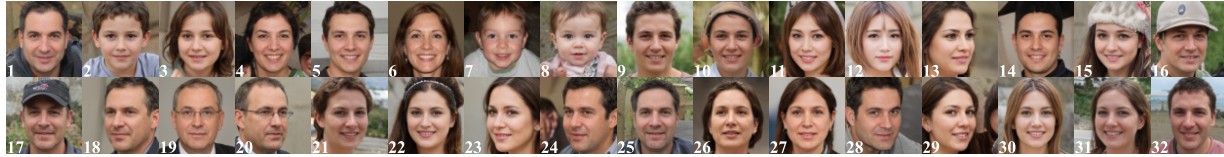

Figure 24: Codebook of a 5 bit SFVQ trained on $\mathcal{W}$ space of StyleGAN2 pre-trained on the FFHQ dataset.

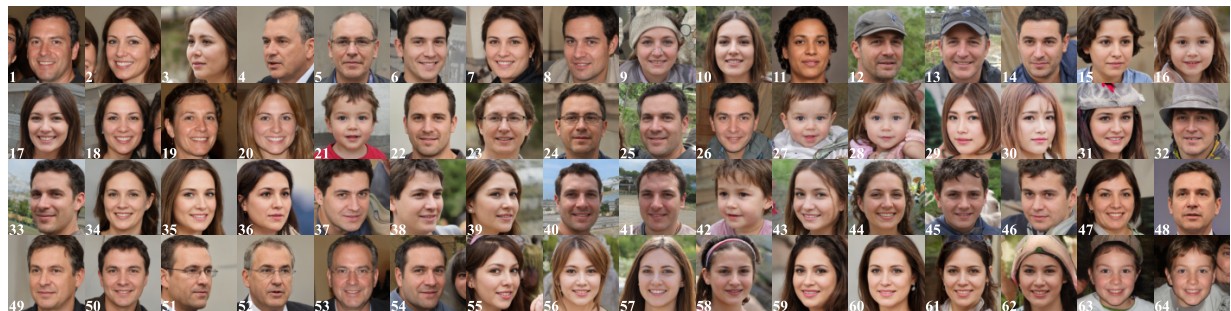

Figure 25: Codebook of a 6 bit SFVQ trained on $\mathcal{W}$ space of StyleGAN2 pre-trained on the FFHQ dataset.

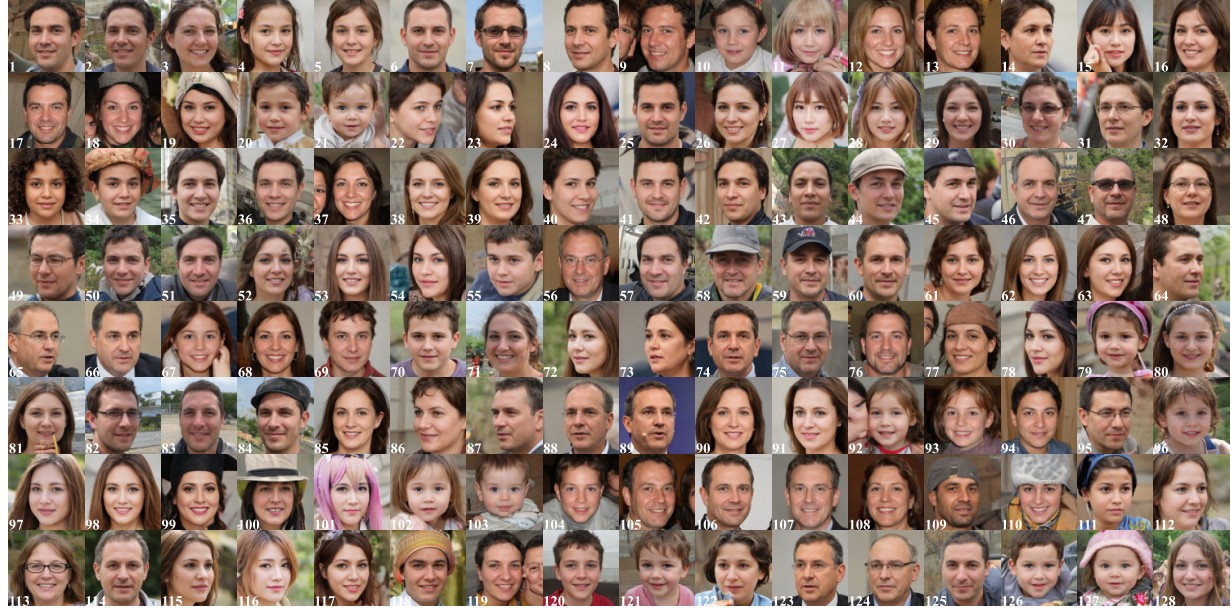

Figure 26: Codebook of a 7 bit SFVQ trained on $\mathcal{W}$ space of StyleGAN2 pre-trained on the FFHQ dataset.

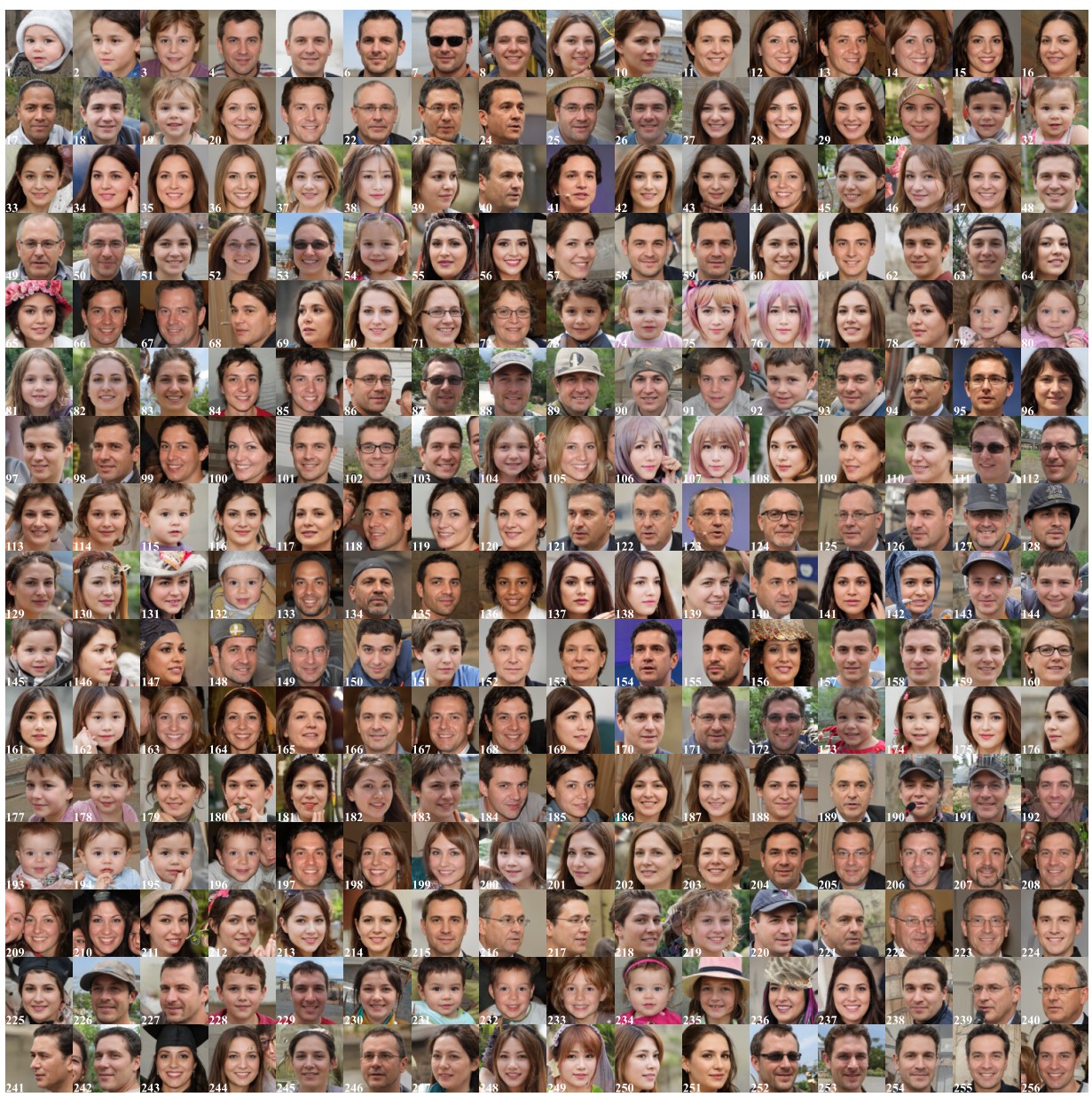

Figure 27: Codebook of a 8 bit SFVQ trained on $\mathcal{W}$ space of StyleGAN2 pre-trained on the FFHQ dataset.

## A.8 SFVQ is not sensitive to training hyper-parameters

As discussed in Sec. 4, training of SFVQ and its interpretation ability is not sensitive to the training hyper-parameters. To prove this claim, we trained the SFVQ on the intermediate latent space ($\mathcal{W}$) of StyleGAN2 pre-trained on the CIFAR10 dataset over different SFVQ bitrates $\{6, 7, 8\}$, batch sizes $\{32, 64, 128\}$, and learning rates $\{5.5e^{-4}, 1e^{-3}\}$. After learning the SFVQ codebook over these different hyper-parameter settings, we plotted the generated images corresponding to the learned codebook in the figures from Fig. 28 to Fig. 45. Apart from the generated images, we also plotted the heatmap of distances between different SFVQ codebook entries. According to all these figures, we see a clear arrangement in the SFVQ codebook over all different settings, as there is an obvious order and distinction between different CIFAR10 data classes, both in the generated images and the heatmaps. In other words, images from an identical data class are organized into groups in both generated images and heatmap plots.

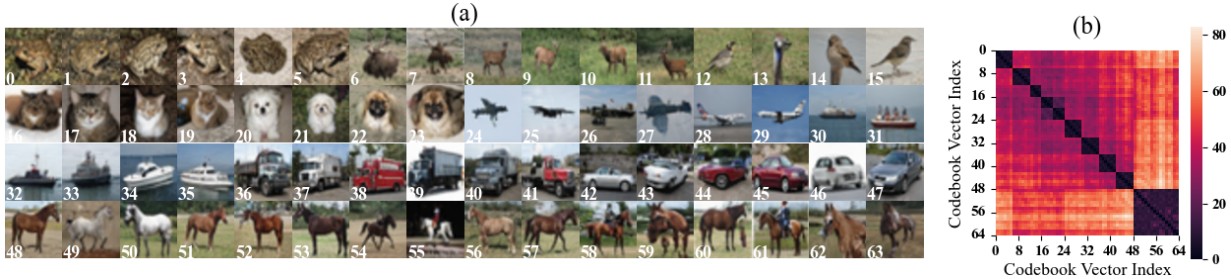

Figure 28: (a) Generated images from codebook of a **6 bit** SFVQ trained on $\mathcal{W}$ space of StyleGAN2 pre-trained on the CIFAR10 dataset when **batch size=32** and **learning rate=5.5e$^{-4}$**. (b) Heatmap of Euclidean distances between all codebook vectors.

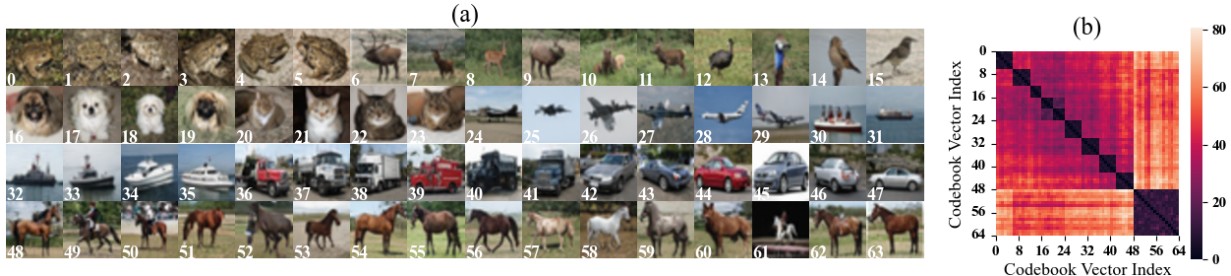

Figure 29: (a) Generated images from codebook of a **6 bit** SFVQ trained on $\mathcal{W}$ space of StyleGAN2 pre-trained on the CIFAR10 dataset when **batch size=64** and **learning rate=5.5e$^{-4}$**. (b) Heatmap of Euclidean distances between all codebook vectors.

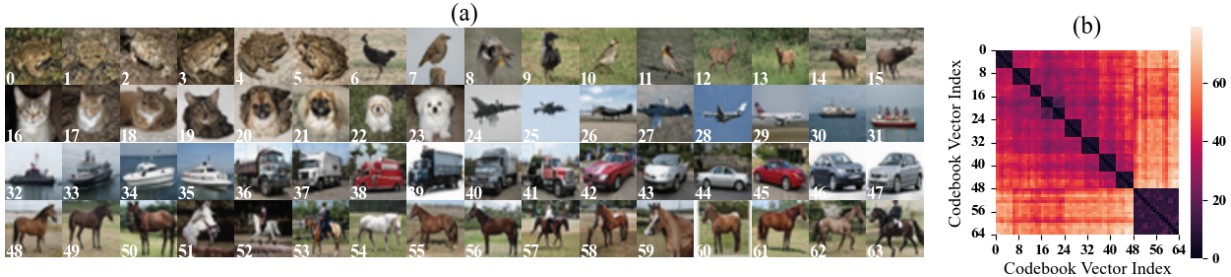

Figure 30: (a) Generated images from codebook of a **6 bit** SFVQ trained on $\mathcal{W}$ space of StyleGAN2 pre-trained on the CIFAR10 dataset when **batch size=128** and **learning rate=5.5e$^{-4}$**. (b) Heatmap of Euclidean distances between all codebook vectors.

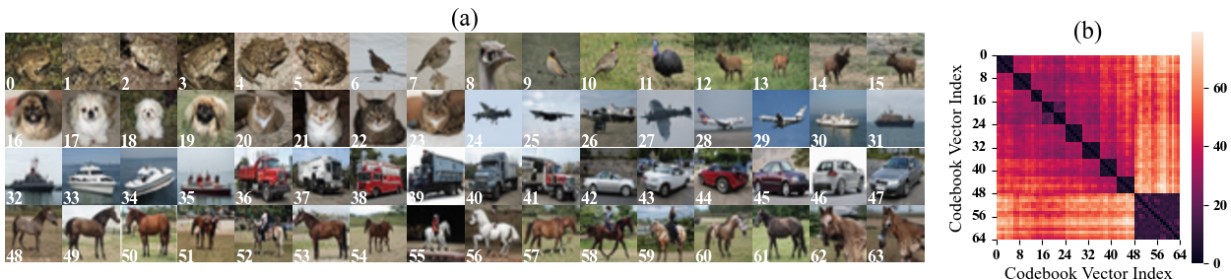

Figure 31: (a) Generated images from codebook of a **6 bit** SFVQ trained on $\mathcal{W}$ space of StyleGAN2 pre-trained on the CIFAR10 dataset when **batch size=32** and **learning rate=1e$^{-3}$**. (b) Heatmap of Euclidean distances between all codebook vectors.

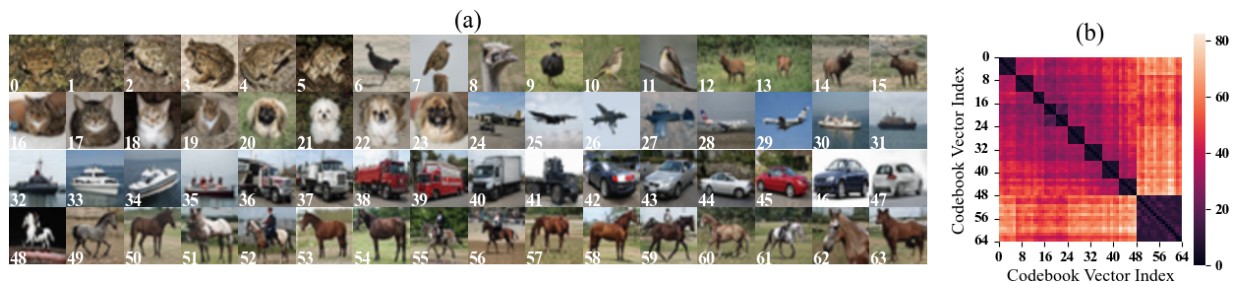

Figure 32: (a) Generated images from codebook of a **6 bit** SFVQ trained on $\mathcal{W}$ space of StyleGAN2 pre-trained on the CIFAR10 dataset when **batch size=64** and **learning rate=1e$^{-3}$**. (b) Heatmap of Euclidean distances between all codebook vectors.

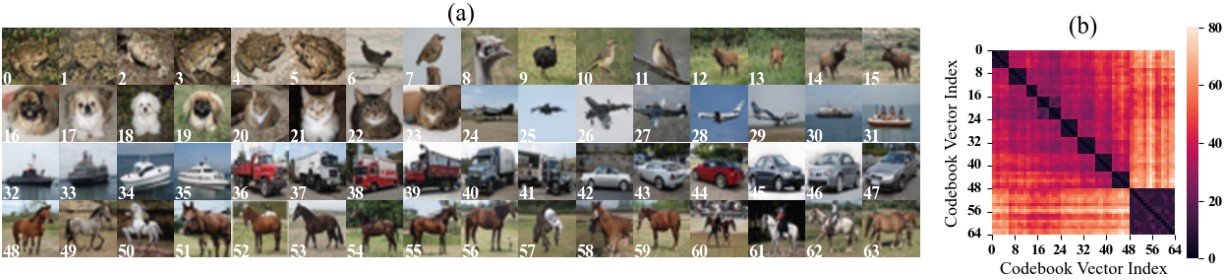

Figure 33: (a) Generated images from codebook of a **6 bit** SFVQ trained on $\mathcal{W}$ space of StyleGAN2 pre-trained on the CIFAR10 dataset when **batch size=128** and **learning rate=1e$^{-3}$**. (b) Heatmap of Euclidean distances between all codebook vectors.

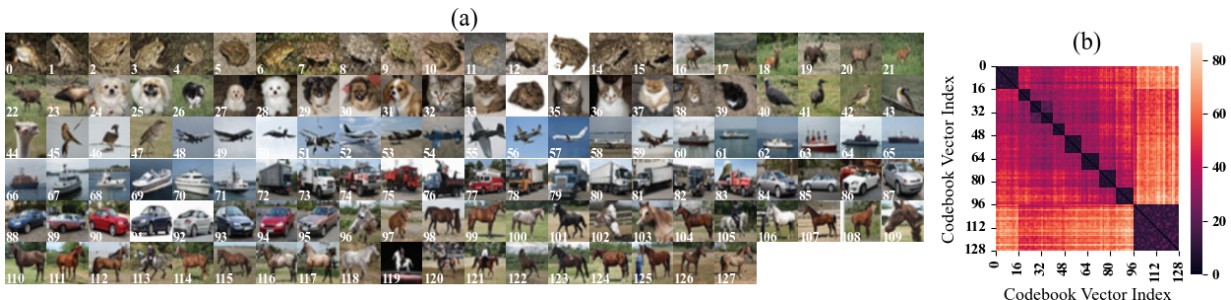

Figure 34: (a) Generated images from codebook of a **7 bit** SFVQ trained on $\mathcal{W}$ space of StyleGAN2 pre-trained on the CIFAR10 dataset when **batch size=32** and **learning rate=5.5e$^{-4}$**. (b) Heatmap of Euclidean distances between all codebook vectors.

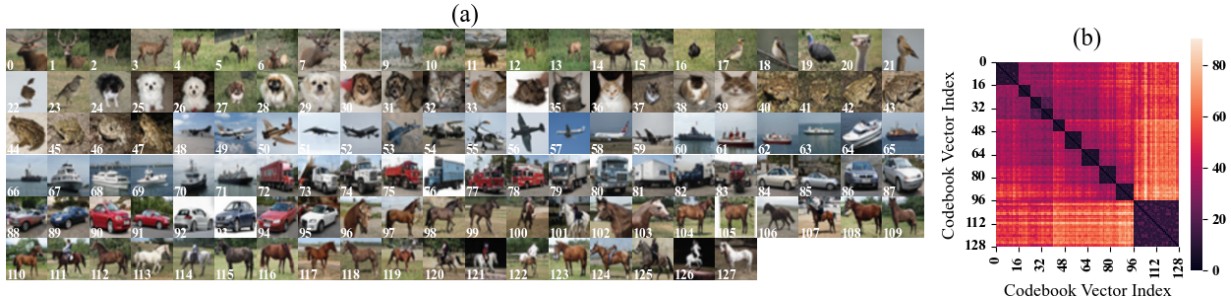

Figure 35: (a) Generated images from codebook of a **7 bit** SFVQ trained on $\mathcal{W}$ space of StyleGAN2 pre-trained on the CIFAR10 dataset when **batch size=64** and **learning rate=5.5e$^{-4}$**. (b) Heatmap of Euclidean distances between all codebook vectors.

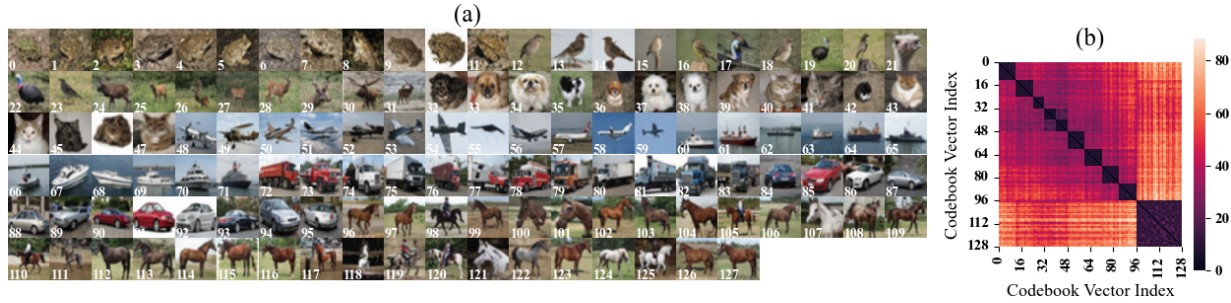

Figure 36: (a) Generated images from codebook of a **7 bit** SFVQ trained on $\mathcal{W}$ space of StyleGAN2 pre-trained on the CIFAR10 dataset when **batch size=128** and **learning rate=5.5e$^{-4}$**. (b) Heatmap of Euclidean distances between all codebook vectors.

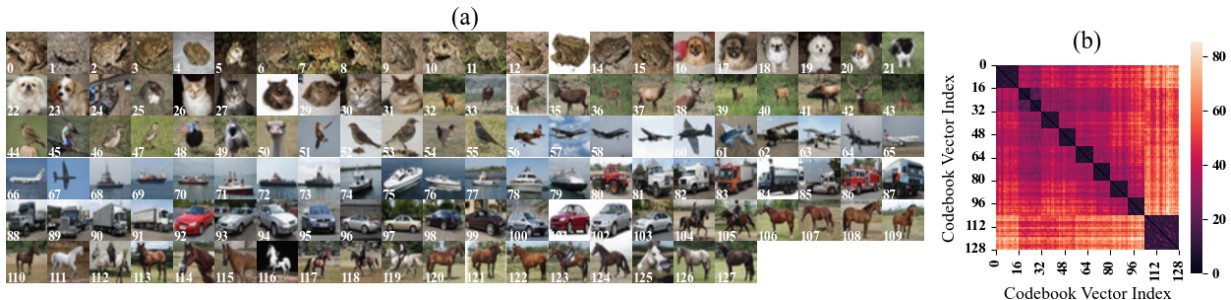

Figure 37: (a) Generated images from codebook of a **7 bit** SFVQ trained on $\mathcal{W}$ space of StyleGAN2 pre-trained on the CIFAR10 dataset when **batch size=32** and **learning rate=1e$^{-3}$**. (b) Heatmap of Euclidean distances between all codebook vectors.

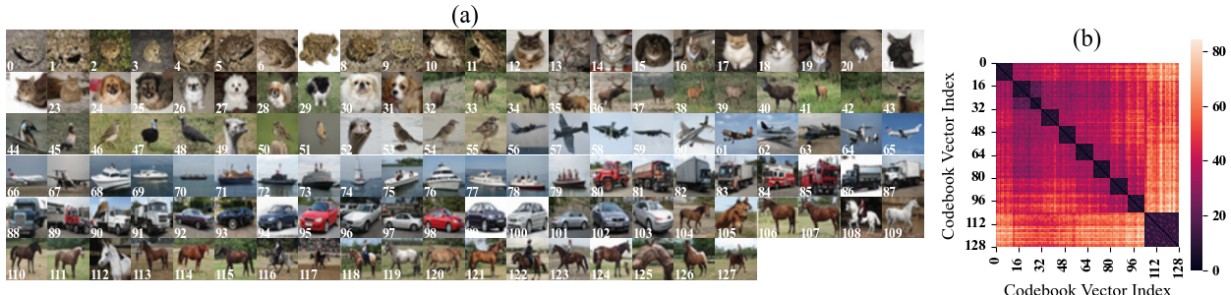

Figure 38: (a) Generated images from codebook of a **7 bit** SFVQ trained on $\mathcal{W}$ space of StyleGAN2 pre-trained on the CIFAR10 dataset when **batch size=64** and **learning rate=1e$^{-3}$**. (b) Heatmap of Euclidean distances between all codebook vectors.

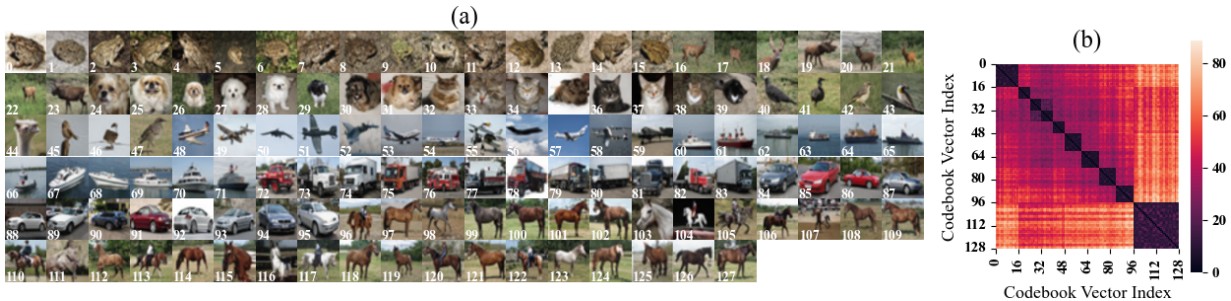

Figure 39: (a) Generated images from codebook of a **7 bit** SFVQ trained on $\mathcal{W}$ space of StyleGAN2 pre-trained on the CIFAR10 dataset when **batch size=128** and **learning rate=1e$^{-3}$**. (b) Heatmap of Euclidean distances between all codebook vectors.

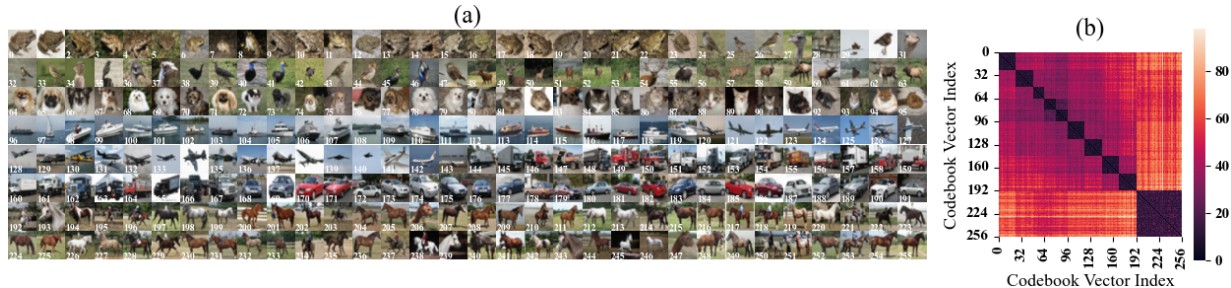

Figure 40: (a) Generated images from codebook of a **8 bit** SFVQ trained on $\mathcal{W}$ space of StyleGAN2 pre-trained on the CIFAR10 dataset when **batch size=32** and **learning rate=5.5e$^{-4}$**. (b) Heatmap of Euclidean distances between all codebook vectors.

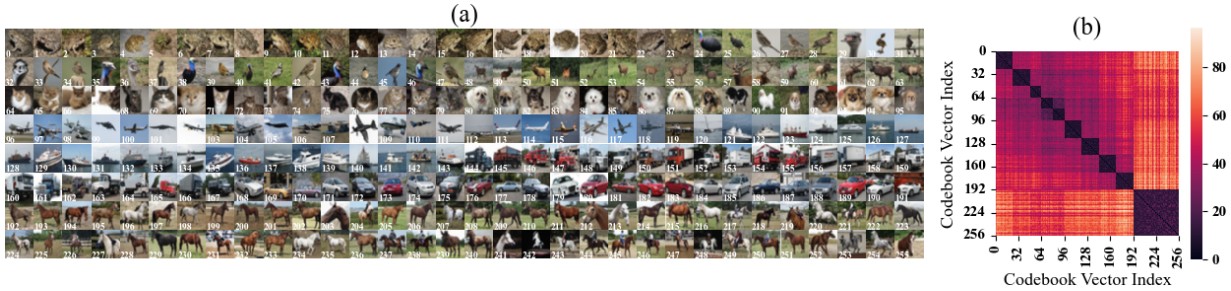

Figure 41: (a) Generated images from codebook of a **8 bit** SFVQ trained on $\mathcal{W}$ space of StyleGAN2 pre-trained on the CIFAR10 dataset when **batch size=64** and **learning rate=5.5e$^{-4}$**. (b) Heatmap of Euclidean distances between all codebook vectors.

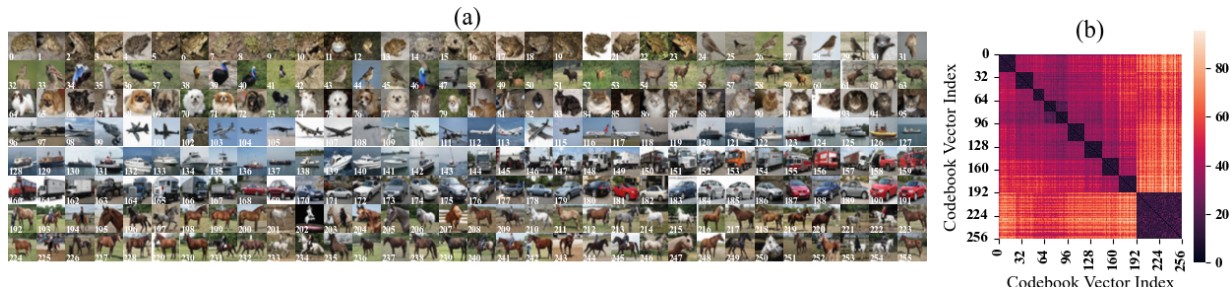

Figure 42: (a) Generated images from codebook of a **8 bit** SFVQ trained on $\mathcal{W}$ space of StyleGAN2 pre-trained on the CIFAR10 dataset when **batch size=128** and **learning rate=5.5e$^{-4}$**. (b) Heatmap of Euclidean distances between all codebook vectors.

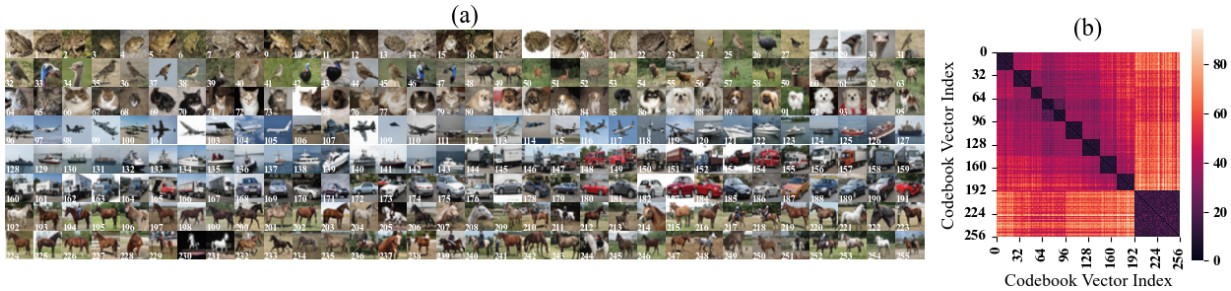

Figure 43: (a) Generated images from codebook of a **8 bit** SFVQ trained on $\mathcal{W}$ space of StyleGAN2 pre-trained on the CIFAR10 dataset when **batch size=32** and **learning rate=1e$^{-3}$**. (b) Heatmap of Euclidean distances between all codebook vectors.

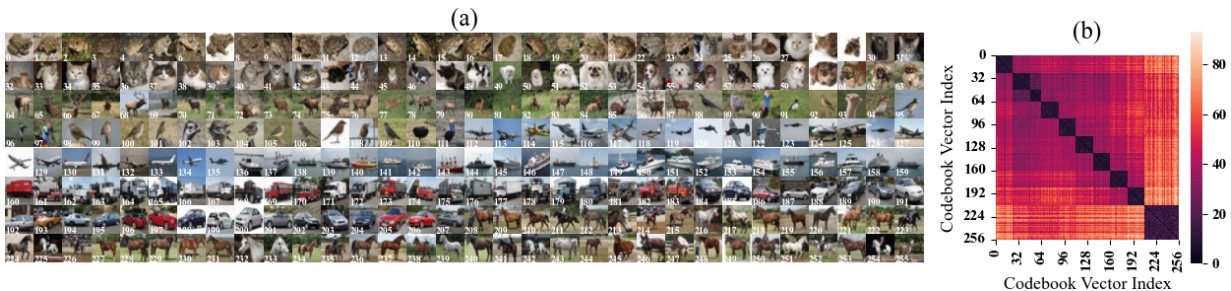

Figure 44: (a) Generated images from codebook of a **8 bit** SFVQ trained on $\mathcal{W}$ space of StyleGAN2 pre-trained on the CIFAR10 dataset when **batch size=64** and **learning rate=1e$^{-3}$**. (b) Heatmap of Euclidean distances between all codebook vectors.

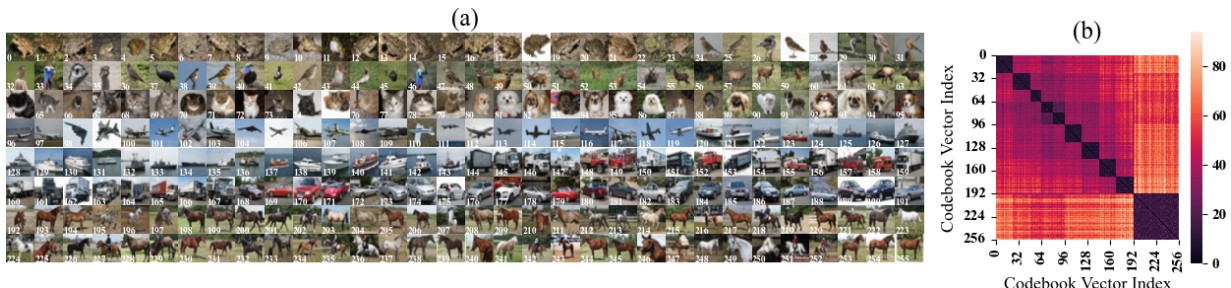

Figure 45: (a) Generated images from codebook of a **8 bit** SFVQ trained on $\mathcal{W}$ space of StyleGAN2 pre-trained on the CIFAR10 dataset when **batch size=128** and **learning rate=1e$^{-3}$**. (b) Heatmap of Euclidean distances between all codebook vectors.

