# *Supplementary Material*

# Unsupervised Panoptic Interpretation of Latent Spaces in GANs Using Space-Filling Vector Quantization

**Mohammad Hassan Vali**                                          *mohammad.vali@aalto.fi*
*Department of Computer Science, Aalto University*

**Tom Bäckström**                                                *tom.backstrom@aalto.fi*
*Department of Information and Communications Engineering, Aalto University*

**Reviewed on OpenReview:** *https: // openreview. net/ forum? id=SEJatSGZX8*

## Subjective comparison with other methods

In this section, we subjectively compare the discovered directions by our proposed method (SFVQ) with GANSpace (Härkönen et al., 2020), LatentCLR (Yüksel et al., 2021) and SeFa (Shen & Zhou, 2021) methods. The figures in the following pages demonstrate the comparisons over 50 different random latent vectors. The readers can generate similar figures for comparison using `demo.py` code in our GitHub demo directory at https://github.com/Speech-Interaction-Technology-Aalto-U/Interpretable-GANs-by-SFVQ.git.

Note that the transformations for the discovered directions are restricted by the dataset bias (Jahanian et al., 2019). Hence, the *Bald* direction is expected to work properly for males, as there are no female faces with bald characteristics in the FFHQ dataset.

## References

Erik Härkönen, Aaron Hertzmann, Jaakko Lehtinen, and Sylvain Paris. Ganspace: Discovering interpretable gan controls. In *Proceedings of NeurIPS*, pp. 9841–9850, 2020.

Ali Jahanian, Lucy Chai, and Phillip Isola. On the" steerability" of generative adversarial networks. In *International Conference on Learning Representations*, 2019.

Yujun Shen and Bolei Zhou. Closed-form factorization of latent semantics in gans. In *Proceedings of the IEEE/CVF conference on computer vision and pattern recognition*, pp. 1532–1540, 2021.

Oğuz Kaan Yüksel, Enis Simsar, Ezgi Gülperi Er, and Pinar Yanardag. Latentclr: A contrastive learning approach for unsupervised discovery of interpretable directions. In *Proceedings of the IEEE/CVF International Conference on Computer Vision*, pp. 14263–14272, 2021.

rotation | $\sigma$=2.67

smile | $\sigma$=2.67

hair_color | $\sigma$=2.67

gender | $\sigma$=2.67

age | $\sigma$=2.67

bald | $\sigma$=2.67

race | $\sigma$=2.67

rotation | σ=2.67

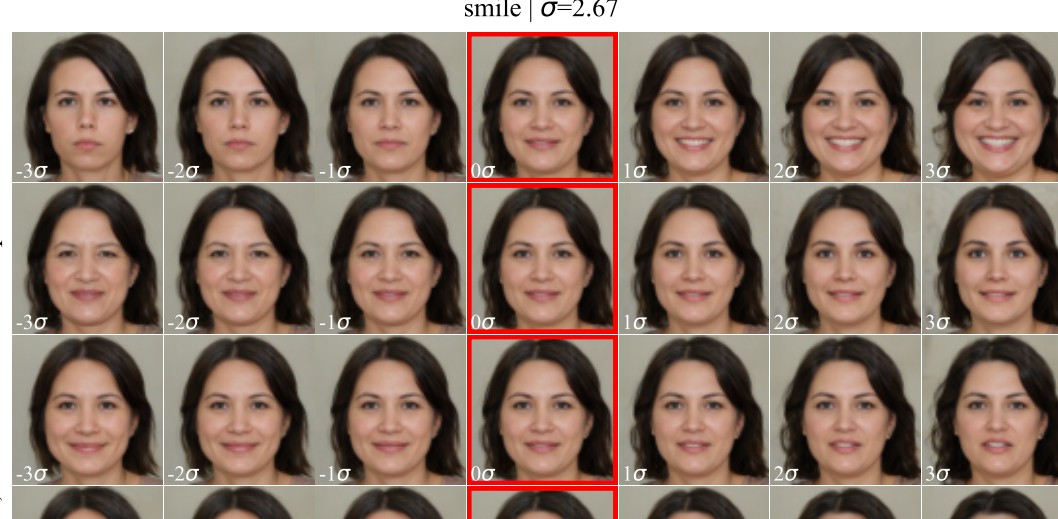

smile | σ=2.67

hair_color | σ=2.67

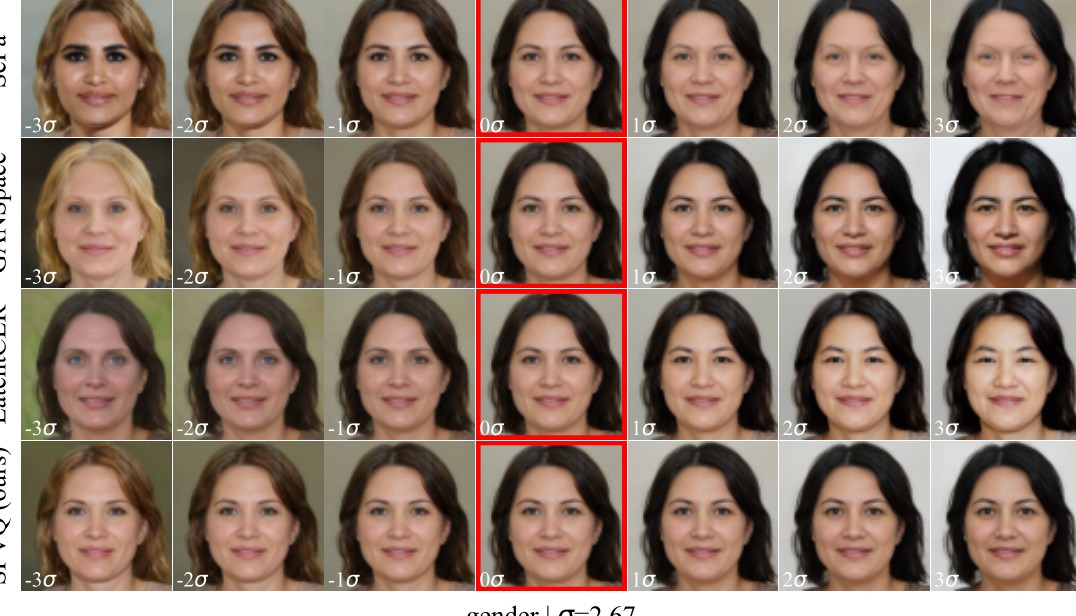

gender | σ=2.67

age | σ=2.67

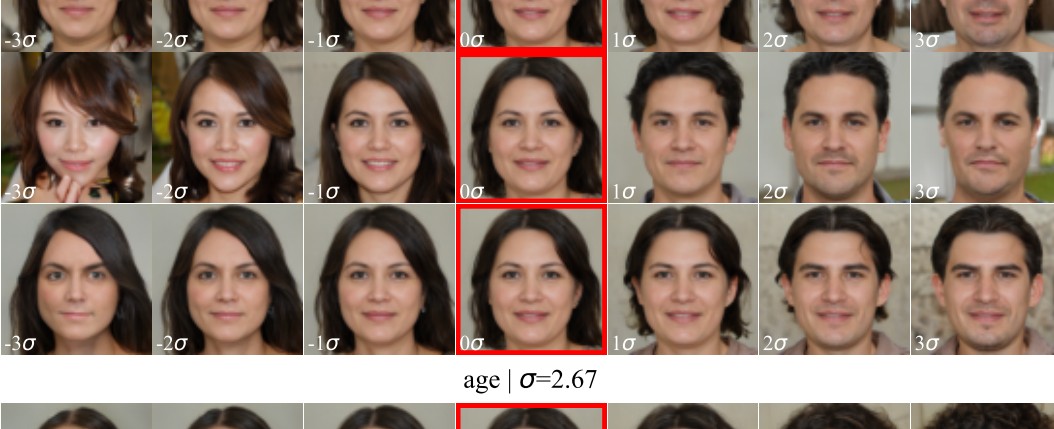

bald | σ=2.67

race | σ=2.67

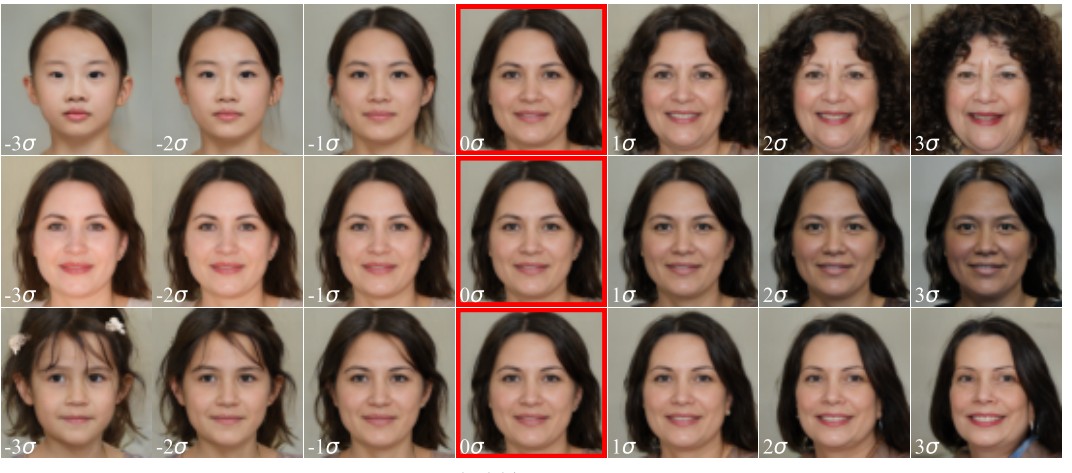

rotation | $\sigma$=2.67

smile | $\sigma$=2.67

hair_color | $\sigma$=2.67

gender | $\sigma$=2.67

age | $\sigma$=2.67

bald | $\sigma$=2.67

race | $\sigma$=2.67

rotation | σ=2.67

smile | σ=2.67

hair_color | σ=2.67

gender | σ=2.67

age | σ=2.67

bald | σ=2.67

race | σ=2.67

rotation | $\sigma$=2.67

smile | $\sigma$=2.67

hair_color | $\sigma$=2.67

gender | $\sigma$=2.67

age | $\sigma$=2.67

bald | $\sigma$=2.67

race | $\sigma$=2.67

rotation | σ=2.67

smile | σ=2.67

hair_color | σ=2.67

gender | σ=2.67

age | σ=2.67

bald | σ=2.67

race | σ=2.67

rotation | $\sigma$=2.67

smile | $\sigma$=2.67

hair_color | $\sigma$=2.67

gender | $\sigma$=2.67

age | $\sigma$=2.67

bald | $\sigma$=2.67

race | $\sigma$=2.67

rotation | σ=2.67

smile | σ=2.67

hair_color | σ=2.67

gender | σ=2.67

age | σ=2.67

bald | σ=2.67

race | σ=2.67

rotation | $\sigma$=2.67

SeFa
GANSpace
LatentCLR
SFVQ (ours)

smile | $\sigma$=2.67

SeFa
GANSpace
LatentCLR
SFVQ (ours)

hair_color | $\sigma$=2.67

SeFa
GANSpace
LatentCLR
SFVQ (ours)

gender | $\sigma$=2.67

SeFa
GANSpace
SFVQ (ours)

age | $\sigma$=2.67

SeFa
LatentCLR
SFVQ (ours)

bald | $\sigma$=2.67

SeFa
LatentCLR
SFVQ (ours)

race | $\sigma$=2.67

SeFa
SFVQ (ours)

rotation | $\sigma$=2.67

smile | $\sigma$=2.67

hair_color | $\sigma$=2.67

gender | $\sigma$=2.67

age | $\sigma$=2.67

bald | $\sigma$=2.67

race | $\sigma$=2.67

rotation | $\sigma$=2.67

smile | $\sigma$=2.67

hair_color | $\sigma$=2.67

gender | $\sigma$=2.67

age | $\sigma$=2.67

bald | $\sigma$=2.67

race | $\sigma$=2.67

rotation | $\sigma$=2.67

smile | $\sigma$=2.67

hair_color | $\sigma$=2.67

gender | $\sigma$=2.67

age | $\sigma$=2.67

bald | $\sigma$=2.67

race | $\sigma$=2.67

rotation | $\sigma$=2.67

smile | $\sigma$=2.67

hair_color | $\sigma$=2.67

gender | $\sigma$=2.67

age | $\sigma$=2.67

bald | $\sigma$=2.67

race | $\sigma$=2.67

rotation | $\sigma$=2.67

smile | $\sigma$=2.67

hair_color | $\sigma$=2.67

gender | $\sigma$=2.67

age | $\sigma$=2.67

bald | $\sigma$=2.67

race | $\sigma$=2.67

rotation | σ=2.67

smile | σ=2.67

hair_color | σ=2.67

gender | σ=2.67

age | σ=2.67

bald | σ=2.67

race | σ=2.67

rotation | $\sigma$=2.67

smile | $\sigma$=2.67

hair_color | $\sigma$=2.67

gender | $\sigma$=2.67

age | $\sigma$=2.67

bald | $\sigma$=2.67

race | $\sigma$=2.67

rotation | $\sigma$=2.67

smile | $\sigma$=2.67

hair_color | $\sigma$=2.67

gender | $\sigma$=2.67

age | $\sigma$=2.67

bald | $\sigma$=2.67

race | $\sigma$=2.67

rotation | $\sigma$=2.67

smile | $\sigma$=2.67

hair_color | $\sigma$=2.67

gender | $\sigma$=2.67

age | $\sigma$=2.67

bald | $\sigma$=2.67

race | $\sigma$=2.67

rotation | σ=2.67

smile | σ=2.67

hair_color | σ=2.67

gender | σ=2.67

age | σ=2.67

bald | σ=2.67

race | σ=2.67

rotation | $\sigma$=2.67

smile | $\sigma$=2.67

hair_color | $\sigma$=2.67

gender | $\sigma$=2.67

age | $\sigma$=2.67

bald | $\sigma$=2.67

race | $\sigma$=2.67

rotation | σ=2.67

smile | σ=2.67

hair_color | σ=2.67

gender | σ=2.67

age | σ=2.67

bald | σ=2.67

race | σ=2.67

rotation | σ=2.67

smile | σ=2.67

hair_color | σ=2.67

gender | σ=2.67

age | σ=2.67

bald | σ=2.67

race | σ=2.67

## rotation | σ=2.67

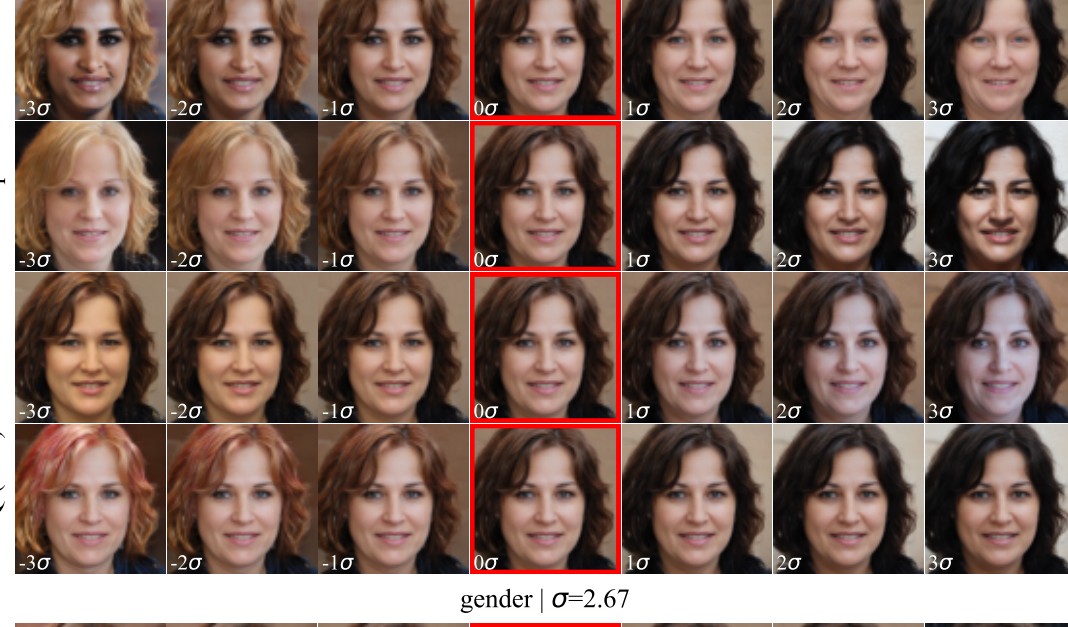

## smile | σ=2.67

## hair_color | σ=2.67

## gender | σ=2.67

## age | σ=2.67

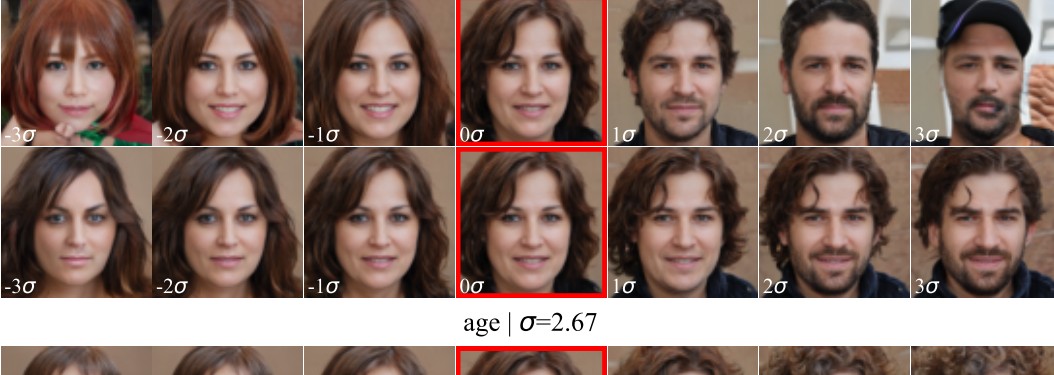

## bald | σ=2.67

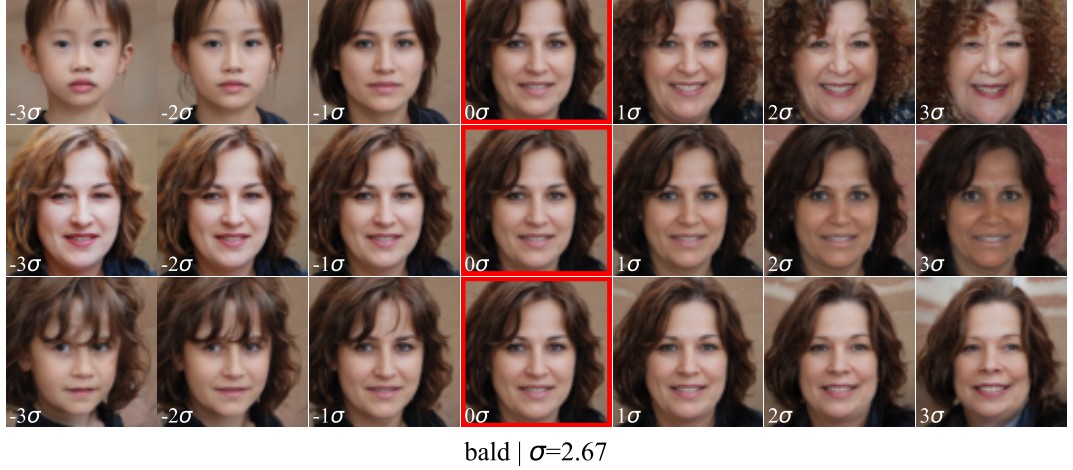

## race | σ=2.67

rotation | σ=2.67

smile | σ=2.67

hair_color | σ=2.67

gender | σ=2.67

age | σ=2.67

bald | σ=2.67

race | σ=2.67

rotation | σ=2.67

smile | σ=2.67

hair_color | σ=2.67

gender | σ=2.67

age | σ=2.67

bald | σ=2.67

race | σ=2.67

rotation | $\sigma$=2.67

smile | $\sigma$=2.67

hair_color | $\sigma$=2.67

gender | $\sigma$=2.67

age | $\sigma$=2.67

bald | $\sigma$=2.67

race | $\sigma$=2.67

rotation | σ=2.67

smile | σ=2.67

hair_color | σ=2.67

gender | σ=2.67

age | σ=2.67

bald | σ=2.67

race | σ=2.67

rotation | $\sigma$=2.67

smile | $\sigma$=2.67

hair_color | $\sigma$=2.67

gender | $\sigma$=2.67

age | $\sigma$=2.67

bald | $\sigma$=2.67

race | $\sigma$=2.67

rotation | σ=2.67

smile | σ=2.67

hair_color | σ=2.67

gender | σ=2.67

age | σ=2.67

bald | σ=2.67

race | σ=2.67

rotation | σ=2.67

smile | σ=2.67

hair_color | σ=2.67

gender | σ=2.67

age | σ=2.67

bald | σ=2.67

race | σ=2.67

rotation | σ=2.67

smile | σ=2.67

hair_color | σ=2.67

gender | σ=2.67

age | σ=2.67

bald | σ=2.67

race | σ=2.67

rotation | σ=2.67

smile | σ=2.67

hair_color | σ=2.67

gender | σ=2.67

age | σ=2.67

bald | σ=2.67

race | σ=2.67

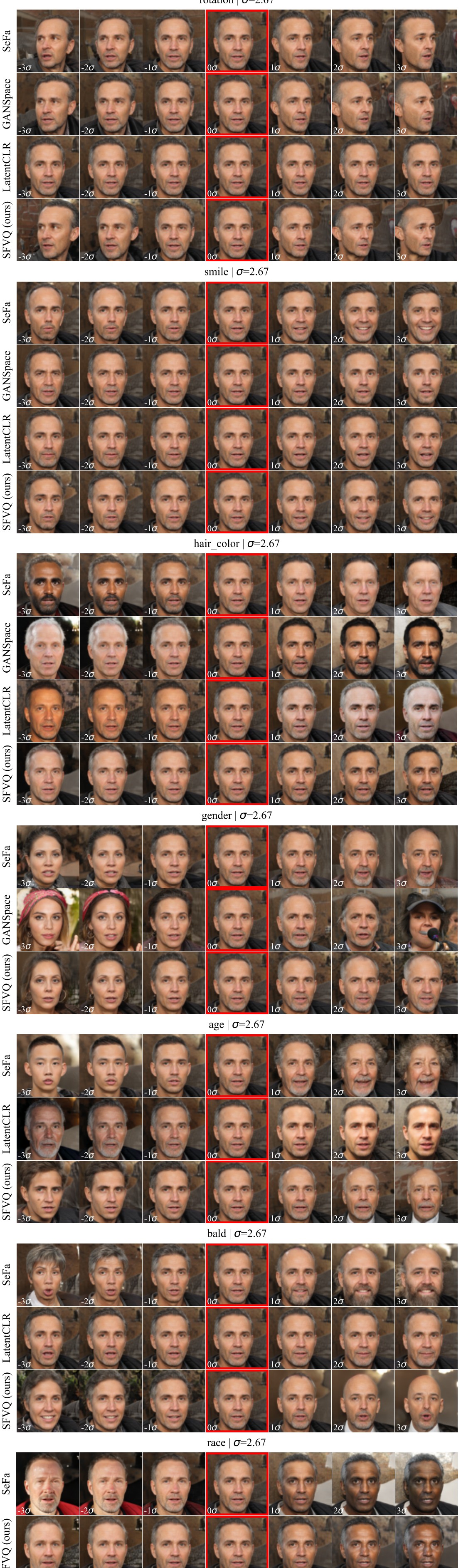

rotation | σ=2.67

smile | σ=2.67

hair_color | σ=2.67

gender | σ=2.67

age | σ=2.67

bald | σ=2.67

race | σ=2.67

rotation | σ=2.67

smile | σ=2.67

hair_color | σ=2.67

gender | σ=2.67

age | σ=2.67

bald | σ=2.67

race | σ=2.67

rotation | $\sigma$=2.67

smile | $\sigma$=2.67

hair_color | $\sigma$=2.67

gender | $\sigma$=2.67

age | $\sigma$=2.67

bald | $\sigma$=2.67

race | $\sigma$=2.67

rotation | $\sigma$=2.67

SeFa · GANSpace · LatentCLR · SFVQ (ours)

$-3\sigma$ $-2\sigma$ $-1\sigma$ $0\sigma$ $1\sigma$ $2\sigma$ $3\sigma$

smile | $\sigma$=2.67

SeFa · GANSpace · LatentCLR · SFVQ (ours)

$-3\sigma$ $-2\sigma$ $-1\sigma$ $0\sigma$ $1\sigma$ $2\sigma$ $3\sigma$

hair_color | $\sigma$=2.67

SeFa · GANSpace · LatentCLR · SFVQ (ours)

$-3\sigma$ $-2\sigma$ $-1\sigma$ $0\sigma$ $1\sigma$ $2\sigma$ $3\sigma$

gender | $\sigma$=2.67

SeFa · GANSpace · SFVQ (ours)

$-3\sigma$ $-2\sigma$ $-1\sigma$ $0\sigma$ $1\sigma$ $2\sigma$ $3\sigma$

age | $\sigma$=2.67

SeFa · LatentCLR · SFVQ (ours)

$-3\sigma$ $-2\sigma$ $-1\sigma$ $0\sigma$ $1\sigma$ $2\sigma$ $3\sigma$

bald | $\sigma$=2.67

SeFa · LatentCLR · SFVQ (ours)

$-3\sigma$ $-2\sigma$ $-1\sigma$ $0\sigma$ $1\sigma$ $2\sigma$ $3\sigma$

race | $\sigma$=2.67

SeFa · SFVQ (ours)

$-3\sigma$ $-2\sigma$ $-1\sigma$ $0\sigma$ $1\sigma$ $2\sigma$ $3\sigma$

rotation | σ=2.67

smile | σ=2.67

hair_color | σ=2.67

gender | σ=2.67

age | σ=2.67

bald | σ=2.67

race | σ=2.67

rotation | $\sigma$=2.67

SeFa
GANSpace
LatentCLR
SFVQ (ours)

$-3\sigma$ $-2\sigma$ $-1\sigma$ $0\sigma$ $1\sigma$ $2\sigma$ $3\sigma$

smile | $\sigma$=2.67

SeFa
GANSpace
LatentCLR
SFVQ (ours)

$-3\sigma$ $-2\sigma$ $-1\sigma$ $0\sigma$ $1\sigma$ $2\sigma$ $3\sigma$

hair_color | $\sigma$=2.67

SeFa
GANSpace
LatentCLR
SFVQ (ours)

$-3\sigma$ $-2\sigma$ $-1\sigma$ $0\sigma$ $1\sigma$ $2\sigma$ $3\sigma$

gender | $\sigma$=2.67

SeFa
GANSpace
SFVQ (ours)

$-3\sigma$ $-2\sigma$ $-1\sigma$ $0\sigma$ $1\sigma$ $2\sigma$ $3\sigma$

age | $\sigma$=2.67

SeFa
LatentCLR
SFVQ (ours)

$-3\sigma$ $-2\sigma$ $-1\sigma$ $0\sigma$ $1\sigma$ $2\sigma$ $3\sigma$

bald | $\sigma$=2.67

SeFa
LatentCLR
SFVQ (ours)

$-3\sigma$ $-2\sigma$ $-1\sigma$ $0\sigma$ $1\sigma$ $2\sigma$ $3\sigma$

race | $\sigma$=2.67

SeFa
SFVQ (ours)

$-3\sigma$ $-2\sigma$ $-1\sigma$ $0\sigma$ $1\sigma$ $2\sigma$ $3\sigma$

rotation | $\sigma$=2.67

smile | $\sigma$=2.67

hair_color | $\sigma$=2.67

gender | $\sigma$=2.67

age | $\sigma$=2.67

bald | $\sigma$=2.67

race | $\sigma$=2.67

rotation | σ=2.67

smile | σ=2.67

hair_color | σ=2.67

gender | σ=2.67

age | σ=2.67

bald | σ=2.67

race | σ=2.67

rotation | $\sigma$=2.67

smile | $\sigma$=2.67

hair_color | $\sigma$=2.67

gender | $\sigma$=2.67

age | $\sigma$=2.67

bald | $\sigma$=2.67

race | $\sigma$=2.67

rotation | $\sigma$=2.67

smile | $\sigma$=2.67

hair_color | $\sigma$=2.67

gender | $\sigma$=2.67

age | $\sigma$=2.67

bald | $\sigma$=2.67

race | $\sigma$=2.67

rotation | $\sigma$=2.67

smile | $\sigma$=2.67

hair_color | $\sigma$=2.67

gender | $\sigma$=2.67

age | $\sigma$=2.67

bald | $\sigma$=2.67

race | $\sigma$=2.67

rotation | $\sigma$=2.67

smile | $\sigma$=2.67

hair_color | $\sigma$=2.67

gender | $\sigma$=2.67

age | $\sigma$=2.67

bald | $\sigma$=2.67

race | $\sigma$=2.67

rotation | σ=2.67

smile | σ=2.67

hair_color | σ=2.67

gender | σ=2.67

age | σ=2.67

bald | σ=2.67

race | σ=2.67

rotation | σ=2.67

smile | σ=2.67

hair_color | σ=2.67

gender | σ=2.67

age | σ=2.67

bald | σ=2.67

race | σ=2.67