# OpenReview forum: "Unsupervised Panoptic Interpretation of Latent Spaces in GANs Using Space-Filling Vector Quantization"
_TMLR — Accepted by TMLR_

### Review · Reviewer_pv2x · 2025-04-03

**Summary Of Contributions:**

The paper considers the problem of learning meaningful directions in the latent space of a GAN. The paper utilizes the space-filling vector quantization (SFVQ) and fits the curve to the space, presenting techniques to improve the training of SFVQ. After the SFVQ is trained, the directions can be read off from the spline. Empirical evidences are provided to demonstrate the effectiveness of the proposed method on pre-trained GANs including StyleGAN2 and BigGAN. It was shown that SFVQ curve provides us with a mechanism to interpret the latent space of the GAN.

**Audience:**

Yes

**Claims And Evidence:**

Yes

**Requested Changes:**

Can the authors elaborate on why the proposed method outperforms the previous ones, despite the model is not explicitly trained to obtain interpretable directions? Can the authors elaborate on why the proposed method is able to yield interpretable latent space in general?

**Strengths And Weaknesses:**

I am not that familiar with the task that the paper is studying.

In terms of strengths, compared with the previous methods for obtaining interpretable directions, the proposed method is conceptually simple, yet able to achieve good results. The SFVQ is also shown to enable a general interpretation of the latent space.

In terms of weaknesses, the paper is largely empirical, without theoretical justifications. I briefly glanced through the previous works on interpretations of GANs' latent space that the authors cited. It appears the previous methods typically involve optimization schemes specifically tailored to obtain interpretable directions, through PCA, contrastive learning, etc. As such, in the current form, it is unclear to me why the proposed method should be able to outperform the previous methods.

---

> ### Author Response · Authors · 2025-05-05
>
> Thanks a lot for your invaluable comments! Here are the answers to the comments. The modifications are highlighted in the paper.
>
> **Weakness and requested change:** In order to present a more clear explanation of the SFVQ method, our modifications to it, and how SFVQ and the proposed modifications work, we made major modifications in Section 3 of the paper (Methods Section) to answer the following question:
>
> * What is the connection between space-filling curves and SFVQ technique, how SFVQ is trained, why SFVQ's codebook has an inherent arrangement and can capture the latent space structure, and why adjacent SFVQ codebook vectors refer to similar contents of the latent space? (**please see Section 3.1**)
> * What are the theoretical justifications of the proposed *initialization* and *codebook expansion* techniques and their practical impact, why our proposed initialization technique works better than random initialization used in the original SFVQ paper [1], and how our proposed initialization and codebook expansion lead to no outlier codebook vectors contrary to [1]? (**please see Sections 3.2.1 and 3.2.2**)
> * Why SFVQ lines refer to meaningful interpretable directions? (**please see Section 3.2.3**)
>
> In general, in Section 3 we added four new figures (Fig.2, Fig.3, Fig.4, and Fig.5) to help better understanding of our proposed method and answer to all above-mentioned questions.
>
> [1] Mohammad Hassan Vali and Tom Bäckström. Interpretable latent space using space-filling curves for phonetic analysis in voice conversion. In Proceedings of Interspeech, 2023.

---

### Review · Reviewer_LTZ1 · 2025-04-11

**Summary Of Contributions:**

This paper proposes using a modified Space-Filling Vector Quantization (SFVQ) for interpreting the latent spaces of pre-trained GANs like StyleGAN2 and BigGAN. The core modifications to SFVQ lie in the initialization and codebook expansion strategies. While the overall goal is interesting and the results visually compelling, the paper suffers from several weaknesses that limit its contribution and impact. The key contribution lies in improving SFVQ through changes to its initialization and code expansion process—specifically, by using convex averaging of codebook vectors instead of simple averaging. This refinement allows the SFVQ curve to more effectively capture the morphological structure of the latent space, making it easier to understand and control. Applied to pretrained GAN models like StyleGAN2 and BigGAN, the method provides a general, unsupervised approach for discovering interpretable directions in latent space, supporting intelligible image transformations and controllable data augmentation. Compared to other techniques, the modified SFVQ is claimed to offer better consistency and interpretability, while remaining architecture-agnostic and broadly applicable across data types.

**Audience:**

Yes

**Broader Impact Concerns:**

There are no major concerns regarding the ethical implications of this work. The proposed method focuses on improving the interpretability of GAN latent spaces through a technical modification of the SFVQ algorithm. While enhanced control over generative models can raise general concerns about misuse, the paper does not introduce any new risks beyond those already associated with standard GAN technologies. Therefore, a Broader Impact Statement is not strictly necessary.

**Claims And Evidence:**

Yes

**Requested Changes:**

* Explore more substantial modifications to SFVQ or develop theoretical justifications for the proposed changes. In particular, the motivation behind the current choice of codebook initialization and its effect is missing. Also in regards to codebook expansion, the motivation behing the choice of the individual weights and their actual effect was not considered.
* In multiple places, authors mention that the proposed method keeps the identity of the test image. However, these is no indept analysis of this aspect? In particular, it would be interesting to test if the convex averaging weights are responsible for this? Alternatively, is there an optimal weights for a specific task, while preserving the identitiy of the test image?
* Further development of a theoretical understansing to analyze the properties of the modified SFVQ and its effectiveness in capturing the latent space structure would be useful.
* Instead of relying solely on correlation, explore other metrics for quantifying interpretability, such as mutual information between latent codes and attributes, or disentanglement scores. Consider using established benchmarks like the DCI Disentanglement metric.
* Investigate Hyperparameter Sensitivity by performing a thorough sensitivity analysis of the bitrate and training parameters, e.g., why in Figure 3 out of 64 codevectors, the horses are overrepresented?

**Strengths And Weaknesses:**

Strengths:

- The images generated from the SFVQ codebooks offer insightful visualizations of the latent space structure, revealing clear clustering based on semantic attributes.
- The lines connecting SFVQ codebook vectors correspond to interpretable directions for image manipulation, enabling controlled image generation.
- The method identifies joint interpretable directions that affect multiple attributes simultaneously, which is a notable contribution.
- Leveraging the SFVQ curve for data augmentation is a novel and promising approach.

Weaknesses:

- The core modifications to SFVQ appear incremental. Initializing codebook vectors using the mean of sorted latent vectors, and employing a convex combination (e.g., 0.99 and 0.01) for codebook expansion to address outlier issues, are practical but not substantial methodological advancements. The paper relies heavily on prior work and lacks significant theoretical contributions.
- Although qualitative comparisons with existing methods are provided, they are largely subjective and lack rigorous evaluation metrics. The quantitative comparisons, while utilizing standard metrics, are limited in scope and do not fully substantiate the claimed improvements. The reported commutativity error is also of limited value, given the linear nature of the transformations. A more robust evaluation—including user studies for subjective assessments and a broader set of quantitative metrics—would strengthen the analysis.
- The paper would benefit from clearer explanations of the proposed modifications and their practical impact. The connection between space-filling curves and the proposed method should be more thoroughly explained. Additionally, some visualizations, particularly the heatmaps, could be enhanced for better clarity and readability.
- While the paper claims that SFVQ is not sensitive to hyperparameters, this is not convincingly demonstrated. A more thorough sensitivity analysis, especially of the bitrate and other training parameters, would help validate this claim.
- The exclusive focus on StyleGAN2 and BigGAN limits the generalizability of the method. Evaluating its performance on a wider range of GAN architectures would enhance the applicability and relevance of the approach.

---

> ### Author Response · Authors · 2025-05-05
> **Answers to Weaknesses**
>
> Thanks a lot for your invaluable comments! Here are the answers to the comments. The modifications are highlighted in the paper.
>
> **Weakness 1:** Apart from solving the outlier issue, our proposed *initialization* lead to a better SFVQ codebook arrangement (please see section 3.2.1 and Figure 4). Although we used SFVQ technique by only improving its *initialization* and *codebook expansion*, we discovered that SFVQ lines refer to interpretable directions which is a new contribution of our paper that was not explored in the original SFVQ paper [1]. Apart from that, joint interpretable directions (section 5.7) and controllable data augmentation (section 5.8) are the new properties we explored from SFVQ in our paper.
>
> **Weakness 2:** We conducted a subjective test over 100 different random latent vectors and asked 20 different human subjects to sort the methods by answering to this question: "Sort the methods that apply the desired change on the test image convincingly, and simultaneously keep the other attributes of the test image (especially the identity) fixed." The results and discussions are provided in the last paragraph of Section 5.4 and Figure 10. The subjective test clearly shows the superiority of our proposed method in 5 interpretable directions out of 7 directions.
>
> **Weakness 3:** In order to present a more clear explanation of the SFVQ method, our modifications to it, and how SFVQ and the proposed modifications work, we made major modifications in Section 3 of the paper (Methods Section) to answer the following question:
>
> * What is the connection between space-filling curves and SFVQ technique, how SFVQ is trained, why SFVQ's codebook has an inherent arrangement and can capture the latent space structure, and why adjacent SFVQ codebook vectors refer to similar contents of the latent space? (**please see Section 3.1**)
> * What are the theoretical justifications of the proposed *initialization* and *codebook expansion* techniques and their practical impact, why our proposed initialization technique works better than random initialization used in the original SFVQ paper [1], and how our proposed initialization and codebook expansion lead to no outlier codebook vectors contrary to [1]? (**please see Sections 3.2.1 and 3.2.2**)
> * Why SFVQ lines refer to meaningful interpretable directions? (**please see Section 3.2.3**)
>
> In general, in Section 3 we added four new figures (Fig.2, Fig.3, Fig.4, and Fig.5) to help better understanding of our proposed method and answer to all above-mentioned questions.
>
>
> **Weakness 4:** We trained SFVQ over different bitrates {6,7,8}, batch sizes {32,64,128}, and learning rates {0.001, 0.00055} on the W space of StyleGAN2 pretrained on FFHQ dataset, and we provided the results in Section A.8 in the appendix. According to the Figures from Fig.28 to Fig.45, we observe that SFVQ training and its interpretation ability is not sensitive to hyper-parameters. We refer to the appendix A.8 in the second paragraph of Section 4, and in the second paragraph of Section 5.1.
>
> **Weakness 5:** We agree with the reviewer, but unfortunately we had limited time and resources with many requests by reviewers during the rebuttal time. We tried our best during recent two weeks to address as many as comments we can. Apart from that, according to our explanations in Section 3.2.3 and Figure 5, since our proposed SFVQ technique works similarly to PCA-based method of GANSpace[2] but with more degrees of freedom, we can expect SFVQ to be able to find interpretable directions in other GANs as well.
>
> [1] Mohammad Hassan Vali and Tom Bäckström. Interpretable latent space using space-filling curves for phonetic analysis in voice conversion. In Proceedings of Interspeech, 2023.
>
> [2] Erik Härkönen, Aaron Hertzmann, Jaakko Lehtinen, and Sylvain Paris. Ganspace: Discovering interpretable gan controls. In Proceedings of NeurIPS, 2020.

---

> > ### Author Response · Authors · 2025-05-05
> > **Answers to Requested Changes**
> >
> > **Requested change 1:** Please refer to the answer to Weakness 3 above.
> >
> > **Requested change 2:** The last paragraph of Section 5.5 (Quantitative Comparison) and Table 2 of the paper exclusively study the identity preservation ability of the methods over different interpretable directions. We follow the papers of WarpedGANSpace [3] and Deep Curvelinear [4], and used the well-known ArcFace [5] pretrained model to evaluate how well identity is preserved when shifting a test image along an interpretable direction. Apart from that, based on the reviewer's request on subjective assessments, we conducted a subjective test and asked the human subjects to rate the interpretable directions by considering their ability to preserve the identity of the test image while editing it. Please refer to the last paragraph of Section 5.4 and Figure 10 in the paper for subjective assessment.
> >
> > As explained in Section 3.2.2, the convex averaging weights (i.e., the proposed *codebook expansion*) does not have anything to do with identity preservation. It is only responsible for training SFVQ with no outlier codebook vectors.
> >
> > **Requested change 3:** Please refer to the answer to Weakness 3 above.
> >
> > **Requested change 4:** Computing mutual information between directions and attributes is not trivial. The amount of shift in each direction and the measured attributes are continuous variables. Some works bin the variables and pretend it is discrete, but this is sensitive to the binning strategy [6]. K-nearest neighbor based KSG estimator [7] can be used to estimate the mutual information, but we need to select the right number of k neighbors carefully and it can be computationally expensive. Correlation metric, on the other hand, is a direct measure of (linear) dependency between directions and attributes, and is used for quantitative evaluations in [3] and [4] in the literature.
> >
> > DCI [8] is an estimate of disentanglement based on regression models with matrices of relative importance. We agree that computing disentanglement (the degree each direction changes one attribute), completeness (the degree each attribute is captured by one direction), and Informativeness (modeled by prediction error) metrics can further evaluate different methods. However, the attributes are measured based on pre-trained models which are biased. Also, these metrics are vulnerable to the choice of regression model [9].
> > In addition to that, we had limited time and resources with many requests by reviewers during the rebuttal time.
> >
> > **Requested change 5:** According to the answer to Weakness 4 above, we have shown in Section A.8 in the appendix that training SFVQ and its interpretation ability is not sensitive to hyper-parameter tuning.
> >
> > Regarding the over-representation of horse class, please refer to the discussions in the second paragraph of Section 5.1, and accordingly the Apeendix A.1 and the figures of Fig.14, Fig.15, Fig.16. According to Figure 14, we observe that the latent vectors of the horse class have the highest diversity in the latent space. That's why the horse class is over-represented. However, if we initialize the SFVQ with N={8,16} initial codewords, the horse class will not be over-represented.
> >
> >
> > [3] Christos Tzelepis, Georgios Tzimiropoulos, and Ioannis Patras. Warpedganspace: Finding non-linear rbf paths in gan latent space. In Proceedings of the IEEE/CVF International Conference on Computer Vision,
> > pp. 6393–6402, 2021.
> >
> > [4] Takehiro Aoshima and Takashi Matsubara. Deep curvilinear editing: Commutative and nonlinear image manipulation for pretrained deep generative model. In Proceedings of the IEEE/CVF Conference on Computer Vision and Pattern Recognition, pp. 5957–5967, 2023.
> >
> > [5] Jiankang Deng, Jia Guo, Niannan Xue, and Stefanos Zafeiriou. Arcface: Additive angular margin loss for deep face recognition. In Proceedings of the IEEE/CVF conference on computer vision and pattern recognition, pp. 4690–4699, 2019.
> >
> > [6] Marc-André Carbonneau, Julian Zaidi, Jonathan Boilard, and Ghyslain Gagnon. Measuring disentanglement: a review of metrics. IEEE Transactions on Neural Networks and Learning Systems, 2022.
> >
> > [7] Weihao Gao, Sewoong Oh, and Pramod Viswanath. Demystifying fixed k-nearest neighbor information estimators. IEEE Transactions on Information Theory, 64(8):5629–5661, 2018.
> >
> > [8] Cian Eastwood and Christopher KI Williams. A framework for the quantitative evaluation of disentangled representations. In International Conference on Learning Representations, 2018.
> >
> > [9] Hsu, K., Dorrell, W., Whittington, J., Wu, J., Finn, C. Disentanglement via latent quantization. Advances in Neural Information Processing Systems, 36, 2023.

---

### Review · Reviewer_JnQ1 · 2025-04-21

**Summary Of Contributions:**

This paper presents a novel approach to interpreting the latent spaces of GANs using space-filling vector quantization (SFVQ). The proposed method introduces a modification to traditional vector quantization that captures the underlying structure of the latent space, making it more interpretable. Experimental results on pretrained StyleGAN2 and BigGAN models across various datasets demonstrate the effectiveness of the proposed method.

**Audience:**

Yes

**Broader Impact Concerns:**

The paper does not have any apparent ethical issues.

**Claims And Evidence:**

Yes

**Requested Changes:**

Please address the issues raised in **Weaknesses**.

**Strengths And Weaknesses:**

**Strengths:**

1. This paper aims to interpret the latent spaces of GANs, which is an interesting and challenging problem.
2. It seems that the proposed method achieves superior results than previous methods.
3. Both qualitative and quantitative experiments have been conducted to evaluate the performance of the proposed method.

**Weaknesses:**

1. The related works and baselines presented in this paper are all from before 2023. Has there been no further research on this topic in recent years?

2. The authors chose StyleGAN2 and BigGAN as the backbones to evaluate the performance of their proposed method. However, StyleGAN2 and BigGAN are from 2020 and 2018, respectively, and are outdated to some extent. Why not experiment on more state-of-the-art GAN models?

3. Currently, diffusion models have significantly surpassed GANs in generative capabilities. Can the method proposed in this paper be applied to diffusion models? Is there a way to interpret the latent spaces of diffusion models?

4. This paper does not include a related work section, which results in an insufficient and unclear presentation of the development and summary of related research.

5. This paper is not well-organized and needs further improvement and refinement.

---

> ### Author Response · Authors · 2025-05-05
>
> Thanks a lot for your invaluable comments! Here are the answers to the comments. The modifications are highlighted in the paper.
>
> **Weakness 1:** As discussed in the first paragraph of Section 5.4, the reasons why we chose the GANSpace, SeFa, and LatentCLR methods as the baselines are:
> * Their interpretable directions for StyleGAN2-FFHQ were readily available in their GitHub repositories. Hence, we can have a fair comparison with these methods without needing to reimpelemt their methods that is prone to errors.
> * Since we intend to use pretrained models in the literature to rate the face attributes like smile, age, rotation, and etc. for our quantitative evaluations, we skipped the methods that were not trained on StyleGAN2-FFHQ.
> * As our method is unsupervised, we planned to compare our method only with unsupervised methods.
> * According to Section 3.2.3 and Figure 5 of the paper, since our proposed method works similarly to the PCA-based method of GANSpace [1] and Eigenvector-based method of SeFa [2], we chose them as baselines for comparison.
>
> Apart from these reasons, there are very recent GAN inversion methods in the literature that use GANSpace [1] to find their interpretable directions such as [3,4], which shows that GANSpace technique is still an effective and important method in the literature. However, we did not compare our method with GAN inversion approaches, as we consider these methods as a different category of image editing methods.
>
> **Weakness 2:** GANSpace [1] interprets the latent space of BigGAN and StyleGAN2 models. Because of similarly of our proposed method with PCA-based method of GANSpace, we also chose BigGAN and StyleGAN2 to have comparisons with GANSpace as the first step. I agree with the reviewer that we could, for example, interpret the latent space of StyleGAN3. However, we know all StyleGAN versions are similar to each other by having the intermediate latent space of W, but they have some specific improvements over each other. So, since our proposed SFVQ technique works similarly to PCA-based method of GANSpace but with more degrees of freedom, we can expect SFVQ to be able to find interpretable directions in StyleGAN3 and other GANs as well. In addition, we had limited time and resources with many requests by reviewers during the rebuttal time, so we could not manage to do new experiments on more recent GANs in this short period of time.
>
> **Weakness 3:** As long as we have a distribution, we can apply SFVQ on that distribution and expect it to learn the underlying structure and interpretable directions of that distribution. Because of SFVQ's connection to space-filling curves (see Section 3.1), it can capture the underlying morphological structure of a distribution. Also, according to Section 3.2.3, SFVQ locates the codebook vectors to the corners of the distribution such that their connecting lines lay along the directions that the distribution has the highest variances (see Figure 5). Because of these reasons, we expect and believe that SFVQ can interpret the latent space of diffusion models as well. But, please notice that according to the title of the paper, in this paper our main focus is interpreting the latent space of GAN models.
>
> **Weakness 4:** We wrote and added a related work section to the paper. We tried our best to provide a thorough literature review of the existing techniques that interpret the of latent spaces of deep neural networks, especially GANs. But, we mainly focused on the methods that find interpretable directions for image editing task (the 3rd category in Section 2).
>
> **Weakness 5:** We added the related work section, and also did major modifications in Section 3 of the paper by providing four new figures (Fig.2, Fig.3, Fig.4, and Fig.5) in order to give a more clear explanation of our proposed method and how it works. We hope these modifications would improve the organization of the paper. But, if there is still space for further improvements, we appreciate if the reviewer mentions more specifically what modifications would help to make the paper well-organized. Then, we will follow the guidelines.
>
> [1] Erik Härkönen, Aaron Hertzmann, Jaakko Lehtinen, and Sylvain Paris. Ganspace: Discovering interpretable gan controls. In Proceedings of NeurIPS, pp. 9841–9850, 2020.
>
> [2] Yujun Shen and Bolei Zhou. Closed-form factorization of latent semantics in gans. In Proceedings of the
> IEEE/CVF conference on computer vision and pattern recognition, pp. 1532–1540, 2021.
>
> [3] Kai Katsumata, Duc Minh Vo, Bei Liu, and Hideki Nakayama. Revisiting latent space of gan inversion for robust real image editing. In Proceedings of the IEEE/CVF Winter Conference on Applications of Computer Vision, pp. 5313–5322, 2024.
>
> [4] Vedant Vasant Dere, Amita Shinde, and Prachi Vast. Conditional reiterative high-fidelity gan inversion for image editing. Pattern Recognition, 147:110068, 2024.

---

### Comment · Editors_In_Chief · 2025-07-02

By the authors' request, on July 2, 2025, the EiCs uploaded a revised PDF subsequent to the camera ready verification. This version differs only in that the section number for Acknowledgments is removed.

---

### Decision · Action_Editor_UXHu · 2025-06-02

**Recommendation:** Accept with minor revision

**Additional Comments:**

*** Decision Justification ***
Two of the reviewers lean towards accept, while a third leans toward reject.  The reviewers generally feel that the clarity of the paper is much improved by updated during the discussion period.  Further, they feel that the paper offers methods for interpretable image changes that are empirically shown to be effective on the tested models.  The reviewer that leans towards reject does so mainly on the basis of limited contribution and recommends further work to expand experiments or add additional theoretical justification.  On review, the AE feels that the acceptance criteria of TMLR will be met if the authors do a minor revision to improve the specificity and clarity of the paper's claims, as indicated below.

*** Requested revisions ***
In addition to the changes made during the reviewer discussion:

The introduction ends with a paragraph describing the main contributions, followed a bulleted list of contributions.  The bullets are not easy interpret / validate on their own. For instance, the paragraph specifies the claims refer to StyleGAN models, but the list of contributions does not specify the range of applicability that is demonstrated. The authors are recommended to merge the bullets and the paragraph to reduce redundancy and improve clarity and to ensure that the claims are specific enough to be verifiable.

Also, note that the paper should generally use the present tense, e.g. "In this paper, we use a modification", rather than "we used".

**Audience:**

Yes

**Audience Explanation:**

Controllable image generation is still of broad interest, and this paper may be of interest to researchers in that area.  The interest may be limited, however, in that latent diffusion models have overtaken GANs for many domains of generation, and, as noted by reviewers the paper offers limited novelty and theoretical contribution.

**Claims And Evidence:**

Yes

**Claims Explanation:**

While broader experiments could be beneficial, as suggested by LTZ1, the experiments adequately support the claims regarding the use of SFVQ to find interpretable directions of change in StyleGAN and similar models.